# Stochastic reconstruction of spatio-temporal rainfall pattern by inverse hydrologic modeling

Jens Grundmann[1], Sebastian Hörning[2], and András Bárdossy[3]

[1]Technische Universität Dresden, Institute of Hydrology and Meteorology, Dresden, Germany
[2]University of Queensland, School of Earth and Environmental Sciences, Brisbane, Australia
[3]Universität Stuttgart, Institute for Modelling Hydraulic and Environmental Systems, Stuttgart, Germany

**Correspondence:** (jens.grundmann@tu-dresden.de)

**Abstract.** Knowledge of spatio-temporal rainfall patterns is required as input for distributed hydrologic models used for tasks such as flood runoff estimation and modeling. Normally, these patterns are generated from point observations on the ground using spatial interpolation methods. However, such methods fail in reproducing the true spatio-temporal rainfall pattern, especially in data scarce regions with poorly gauged catchments, or for highly dynamic, small scaled rainstorms which are not well recorded by existing monitoring networks. Consequently, uncertainties arise in distributed rainfall-runoff modeling if poorly identified spatio-temporal rainfall pattern are used, since the amount of rainfall received by a catchment as well as the dynamics of the runoff generation of flood waves are underestimated. To address this problem we propose an inverse hydrologic modeling approach for stochastic reconstruction of spatio-temporal rainfall pattern. The methodology combines the stochastic random field simulator Random Mixing and a distributed rainfall-runoff model in a Monte-Carlo framework. The simulated spatio-temporal rainfall patterns are conditioned on point rainfall data from ground monitoring networks and the observed hydrograph at the catchment outlet and aim to explain measured data at best. Since we infer from an integral catchment response on a three-dimensional input variable, several candidates of spatio-temporal rainfall pattern are feasible and allow for an analysis of their uncertainty. The methodology is tested on a synthetic rainfall-runoff event on sub-daily time steps and spatial resolution of 1km² for a catchment covered by rainfall partly. A set of plausible spatio-temporal rainfall patterns can be obtained by applying this inverse approach. Furthermore, results of a real world study for a flash flood event in a mountainous arid region are presented. They underline that knowledge about the spatio-temporal rainfall pattern is crucial for flash flood modeling even in small catchments and arid and semiarid environments.

## 1   Motivation

The importance of spatio-temporal rainfall pattern for rainfall-runoff estimation and modeling is well known in hydrology, and has been addressed by several simulation studies, especially since distributed hydrologic models have become available. Many of those studies demonstrated the effect of resulting runoff responses for different spatial rainfall pattern (Beven and Hornberger, 1982; Obled et al., 1994; Morin et al., 2006; Nicotina et al., 2008), or addressed the errors in runoff prediction and the difficulties in parameterisation and calibration of hydrologic models if the spatial distribution of rainfall is not well known (Troutman, 1983; Lopes, 1996; Chaubey et al., 1999; Andreassian et al., 2001). As a consequence, studies were performed

to investigate configurations of rainfall monitoring networks (Faures et al., 1995), and rainfall errors and uncertainties for hydrologic modeling (McMillan et al., 2011; Renard et al., 2011).

In general, rainfall monitoring networks based on point observations on the ground (station data) require interpolation methods to obtain spatio-temporal rainfall fields usable for distributed hydrologic modeling. Traditional interpolation methods fail in reproducing the true spatio-temporal rainfall pattern especially for: (i) data scarce regions with poorly gauged catchments and low network density, (ii) highly dynamic, small scaled rainstorms which are not well recorded by existing monitoring networks, and (iii) catchments which are covered by rainfall partly. Consequently, uncertainties are associated with poorly identified spatio-temporal rainfall pattern in distributed rainfall-runoff-modeling since the amount of rainfall received by a catchment as well as the dynamics of runoff generation processes are typically underestimated by current methods.

The effects of poorly estimated spatio-temporal rainfall fields are visible in particular for semiarid and arid regions, where rainstorms show a great variability in space and time and the density of ground monitoring networks is sparsely compared to other regions (Pilgrim et al., 1988). Based on an analysis of 36 events in a mountainous region of Oman, McIntyre et al. (2007) show a wide range of event based runoff coefficients, which underlines that achieving reliable runoff predictions by using hydrologic models in those regions is extremely challenging. This is supported by several simulation studies (Al-Qurashi et al., 2008; Bahat et al., 2009), who address the uncertainties in model parameterisation due to uncertain rainfall input. In this context Gunkel and Lange (2012) report that reliable model parameter estimation was only possible by using rainfall rader. However, this information is not available everywhere.

To address the inherent uncertainties described above, stochastic rainfall generators are used intensively to create spatio-temporal rainfall inputs for distributed hydrologic models to transform rainfall into runoff. A large amount of literature exists describing different approaches for space-time simulation of rainfall fields; among them multi-site temporal simulation frameworks (Wilks, 1998), approaches based on the theory of random fields (Bell, 1987; Pegram and Clothier, 2001) or approaches based on the theory of point processes and its generalization, which includes the popular Turning bands method (Mantoglou and Wilson, 1982). Enhancements were made in order to portray different rain storm pattern and distinct properties of rainfall fields, like spatial covariance structure, space-time anomaly, and intermittency (see Leblois and Creutin 2013; Paschalis et al. 2013; Peleg et al. 2017).

Applications of spatio-temporal rainfall simulations together with hydrologic models are of straightforward, Monte-Carlo type, where a large number of potential rainfall fields are generated driven by stochastic properties of observed rainstorms or longer time series. These fields are used as inputs for distributed hydrologic model simulations to investigate the impact of certain aspects of rainfall like uncertainty in measured rain depth, spatial variability, etc., on simulated catchment responses. Rainfall simulation applications are performed in unconditional mode (reproducing rain field statistics only) or conditional mode, where observations (e.g from rain gauges) are reproduced too. The latter are commonly used for investigating the effect of spatial variability using fixed total precipitation and variations in spatial pattern (Krajewski et al., 1991; Shah et al., 1996; Casper et al., 2009; Paschalis et al., 2014). However, stochastic rainfall simulations in combination with distributed hydrologic modeling can be computationally demanding and can fail at matching the observed stream flow if rainfall fields are conditioned on rainfall point observations only.

On the other hand, inverse hydrologic modeling approaches have been developed to estimate rainfall time series based on observed stream flow data. Those approaches require either an inverting of the underlying mathematical equations for the nonlinear transfer function (Kirchner, 2009; Kretzschmar et al., 2014) or an application of the hydrologic model in a Bayesian inference scheme (Kavetski et al., 2006; Del Giudice et al., 2016). Up to now, both approaches delivers time series of catchment-averaged rainfall only, which gives no idea about the spatial extent and distribution of rainfall. This is particularly important when considering events such as localised rainstorms, which might be underestimated and not accurately portrayed.

The goal here is an event based reconstruction of spatio-temporal rainfall pattern which explain measured point rainfall data and catchment runoff response at best. For that we are looking for potential candidates of rainfall fields for sub daily time steps and spatial resolution of 1km² which, to our knowledge hasn't been done so far. To achieve this task, we combine stochastic rainfall simulations and distributed hydrologic modeling in an inverse modeling approach, where spatio-temporal rainfall pattern are conditioned on rainfall point observations and observed runoff. The methodology of the inverse hydrologic modeling approach consists of the stochastic random field simulator Random Mixing and a distributed rainfall-runoff model in a Monte-Carlo framework. Until now, Random Mixing, developed by Bárdossy and Hörning (2016b) for solving inverse groundwater modeling problems, has been used by Haese et al. (2017) for reconstruction and interpolation of precipitation fields using different data sources for rainfall.

After this introduction the methods are described in section 2. It gives an overview of the methodology and further details for the applied rainfall-runoff model, the Random Mixing and its application for rainfall fields. Section 3 aims to test the methodology. A synthetic test site is introduced which is used to demonstrate and discuss: (i) the limits of common hydrologic modeling approaches (using rainfall interpolation), and (ii) the shortcomings of rainfall simulations which are not conditioned on the observed runoff. In contrast, the functionality of the inverse hydrologic modeling approach is illustrated and discussed. In section 4, the inverse hydrologic modeling approach is applied for real world data by an example of an arid mountainous catchment in Oman. The test site is introduced and results are shown and discussed. Finally, summary and conclusions are given in section 5.

## 2 Methods

### 2.1 General approach

The methodology described here can be characterized as an inverse hydrologic modeling approach. It aims to conclude on potential candidates for the unknown spatio-temporal rainfall pattern based on runoff observations at the catchment outlet, known parameterisation of the rainfall-runoff model and rain gauge observations. The approach combines a grid-based spatially distributed rainfall-runoff model and a conditional random field simulation technique called Random Mixing (Bárdossy and Hörning, 2016a, b). Random Mixing is used to simulate a conditional rainfall field which honors the observed rainfall values as well as their spatial and temporal variability. Afterwards, an optimization is performed to additionally condition the rainfall field on the observed runoff. Therefore, the initial field is used as input to the rainfall-runoff model. The deviation between the simulated runoff and the observed runoff is evaluated based on the model efficiency (NSE) defined by Nash and Sutcliffe

(1970). To minimize this deviation the rainfall field is *mixed* with another random field which exhibits certain properties such that the mixture honors the observed rainfall values and their spatio-temporal variability. This procedure is repeated until a satisfying solution, i.e. a conditional rainfall field that achieves a reasonable NSE, is found. To enable a reasonable uncertainty estimation the procedure is repeated until a predefined number of potential candidates has been found. In the following, rainfall is used interchangeably with precipitation.

## 2.2 Rainfall runoff model

A simple spatially distributed rainfall-runoff (RR) model is used as transfer function to portray the nonlinear transformation of spatially distributed rainfall into runoff at catchment outlets. The model is dedicated to describe rainfall-runoff processes in arid mountainous regions, which are mostly based on infiltration excess and Hortonian overland flow. The model is working on regular grid cells in event based mode. It is parsimonious in number of parameters considering transmission losses but having no base flow component. Pre-state information at the beginning of an event is neglected since runoff processes starting under dry conditions mostly (Pilgrim et al., 1988).

More specifically, only simple approaches known from hydrologic textbooks for the simulation of single rainfall-runoff events (no long-term water balance) are used (Dyck and Peschke, 1983). Effective precipitation $Pe(x,t)$ with location $x \in D$ and time $t \in T$ is calculated by an initial and constant rate loss model applied on each grid cell which is affected by rainfall. The initial loss $I_a(x)$ represents interception and depression storage. If the accumulated precipitation exceeds $I_a(x)$ surface runoff may occur, which is reduce by the constant rate $f_c(x)$ throughout an event to consider infiltration. The calculated effective precipitation (respectively surface runoff) is transferred to the next river channel section considering translation and attenuation processes. Translation is accounted for with a grid-based travel-time function to include the effects of surface slope and roughness. Attenuation is accounted for with a single linear storage unit with recession constant $f_r(x)$. Both approaches are applied on grid cells affected by effective precipitation only to fully support spatial distributed calculations corresponding to the spatial extent of the rain field. The properties of several landscape units are addressed by different parameter sets (for $I_a(x), f_c(x), f_r(x)$) following the concept of hydrogeological response units (Gerner, 2013) (since hydrologic processes are mostly driven by hydrogeology in these regions). Runoff is routed to the catchment outlet by a simple lag model in combination with a constant rate ($f_t$) loss model to portray transmission losses along the stream channel. The RR-model is applied for hourly time step on regular grids cells of 1 km by 1 km. Parameters are assumed to be known and fixed during the inverse modeling procedure. The RR-model is linked to Random Mixing directly and named with working title NAMarid.

## 2.3 Random Mixing for inverse hydrologic modeling

Random Mixing is a geostatistical simulation approach . It uses copulas as spatial random functions (Bárdossy, 2006) and represents an extension to the gradual deformation approach (Hu, 2000). In the following a brief description of the Random Mixing algorithm is presented. A detailed explanation can be found in Hörning (2016).

The goal of the inverse hydrologic modeling approach presented herein is to find a conditional precipitation field $P(x,t)$ with location $x \in D$ and time $t \in T$ which reproduces the observed spatial and temporal variability and marginal distribution

of $P$. This field should also honor precipitation observations at locations $x_j$ and times $t_i$:

$$P(x_j, t_i) = p_{j,i} \quad \text{for} \quad j = 1, \ldots, J \quad \text{and} \quad i = 1, \ldots, I \tag{1}$$

Note that $P$ denotes a spatial field and $p$ denotes a precipitation value within that field. Furthermore, the solution of a rainfall-runoff model using the field $P$ as input variable should approximately honor the observed runoff:

$$Q_t(P) \approx q_t \quad \text{for} \quad t = 1, \ldots, T \tag{2}$$

where $Q_t$ denotes the rainfall-runoff model and $q_t$-s represent the observed runoff values at time step $t$. Note that $Q_t(P)$ represents a non-linear function of the field $P$.

In order to find such a precipitation field $P$ which fulfills the conditions given in Eq. (1) and Eq. (2) Random Mixing can be applied. Figure 1 shows a flowchart of the corresponding procedure.

Using the given observations $p_{j,i}$, a marginal distribution $G(p)$ has to be fitted to them. Note that in general any type of distribution function (e.g. parametric, non-parametric and combinations of distributions) can be used. For the applications presented herein the selected marginal distribution consists of two parts: the discrete probability of zero precipitation and an exponential distribution for the wet precipitation observations. It is defined as:

$$G(p) = \begin{cases} p_0 & \text{if } p = 0 \\ p_0 + p_0(1 - \exp(-\lambda p)) & \text{otherwise} \end{cases} \tag{3}$$

with $p$ denoting precipitation values, $p_0$ is the discrete probability of zero precipitation and $\lambda$ denotes the parameter of the exponential distribution. Thus the parameters that need to be estimated are $p_0$ and $\lambda$. Then, using the fitted marginal distribution the observed precipitation values are transformed to standard normal:

$$w = \begin{cases} < \Phi^{-1}(p_0) & \text{if } p = 0 \\ \Phi^{-1}(p_0 + p_0(1 - \exp(-\lambda p))) & \text{otherwise} \end{cases} \tag{4}$$

where $\Phi^{-1}$ denotes the univariate inverse standard normal distribution. Note that zero precipitation observations are not trans-
formed to the same value, but they are considered as inequality constraints as described in Eq. (4). Thus the spatio-temporal dependence structure of the variable is taken into account as described in Hörning (2016). Further note that the transformation of the marginal distribution described in Eq.4 can be reversed via:

$$P(x, t) = G^{-1}(\Phi(W(x, t))) \tag{5}$$

where $G^{-1}$ denotes the inverse marginal distribution of $P$ and $\Phi$ denotes the univariate standard normal distribution. Also note
that $W$ denotes the transformed spatial field while $w$ denotes a transformed observed value within that field. Note that in this approach we assume that the precipitation distribution is the same for each location $x$ and each time-step $t$. One could use a location and/or time specific distribution to take spatial or temporal non-stationarity into account, however this requires a relatively large amount of precipitation observations and/or additional information.

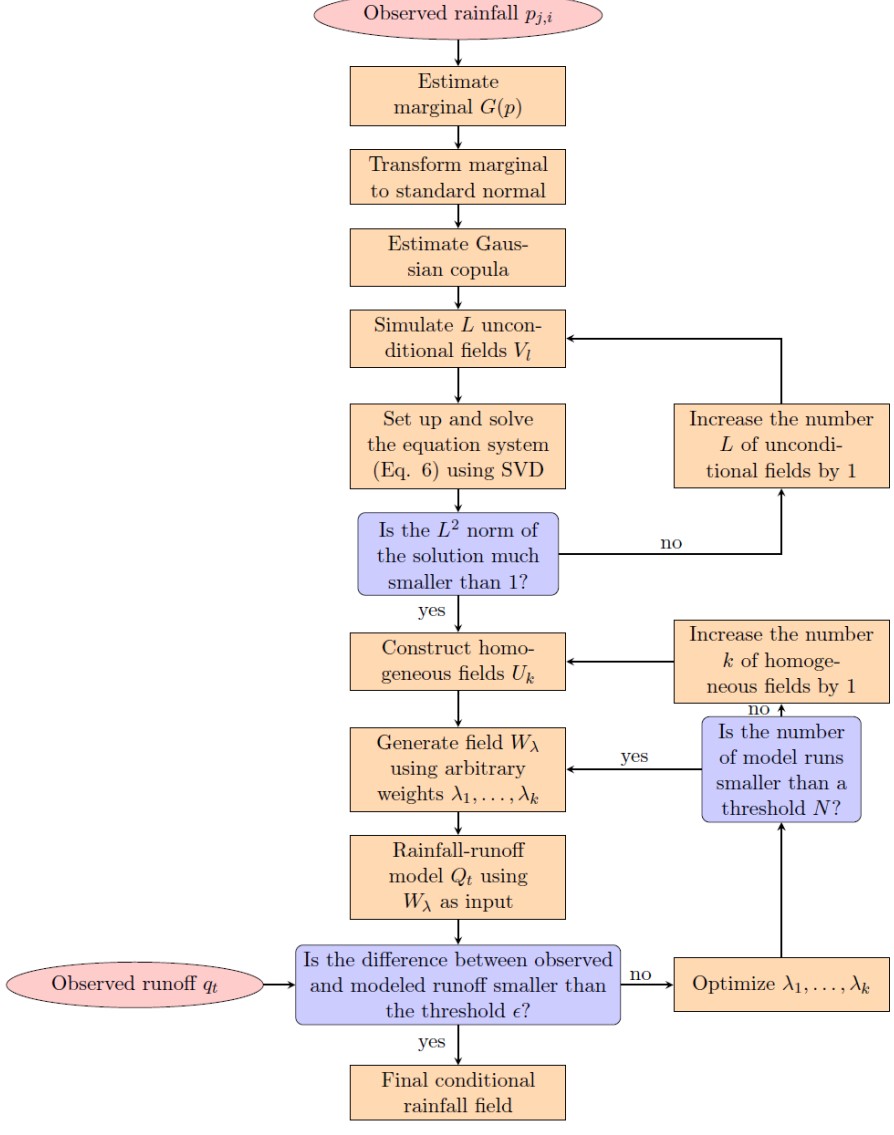

**Figure 1.** Flowchart of the Random Mixing algorithm for inverse hydrologic modeling.

As a next step we assume that the field $W$ is normal, thus its spatio-temporal dependence is described by the normal copula with correlation matrix $\Gamma_c$. In general copulas are multivariate distribution functions defined on the unit hypercube with uniform univariate marginals. They are used to describe the dependence between random variables independently of their marginal distributions. The normal copula can be derived from a multivariate standard normal distribution (see Bárdossy and Hörning (2016b) for details). It enables modeling a Gaussian spatio-temporal dependence structure with arbitrary marginal distribution. Note that its correlation matrix $\Gamma_c$ has to be assessed from the available observations. If no zero observations are present the maximum likelihood estimation procedure described in Li (2010) can be applied to estimate the copula parameters. If zero values are present a modified maximum likelihood approach has to be used (Bárdossy, 2011). It uses a combination of three different cases (wet-wet pairs, wet-dry pairs, dry-dry pairs of observations) for the estimation of the copula parameters.

As a next step, unconditional standard normal random fields $V_l$ with $l = 1, \ldots, L$ are simulated such that they all share the same spatio-temporal dependence structure which is described by $\Gamma_c$ of the fitted normal copula. Such fields can for example be simulated using Fast Fourier Transformation for regular grids (Wood and Chan, 1994; Wood, 1995; Ravalec et al., 2000) or Turning band simulation (Journel, 1974). Here we used the spectral representation method introduced by Shinozuka and Deodatis (1991, 1996). Using the fields $V_l$, the system of linear equations:

$$\sum_{l=1}^{L} \alpha_l V_l(x_j, t_i) = w_{j,i} \quad \text{for} \quad i = 1, \ldots, I \quad j = 1, \ldots, J \quad \text{with} \quad L > N = I \cdot J \tag{6}$$

is set up. Note that $\alpha_l$ denotes the weights of the linear combination, $w_{j,i} = \Phi^{-1}(G(p_{i,j}))$ are the transformed precipitation values and $V_l(x_j, t_i)$ are the values of the random fields at the observation locations. Using singular value decomposition (SVD) (Golub and Kahan, 1965) to solve this equation system leads to a minimum $L^2$ norm solution. In order to obtain a smooth, low variance field a $L^2$ norm $\sum \alpha_l^2 \ll 1$ is required. If no such solution is found, an additional field $V_{L+1}$ is created, added to the system of linear equation and the system is solved again. Note that with increasing degrees of freedom (i.e. more fields) the $L^2$ norm of the solution decreases.

Once a solution with an acceptable $L^2$ norm i.e. $\sum \alpha_l^2 \ll 1$ is found the resulting field is defined as:

$$W^* = \sum_{l=1}^{L+M} \alpha_l V_l \tag{7}$$

where $M$ denotes the number of additional fields added to the equation system. Note that $W^*$ fulfills the conditions defined in Eq. (1) however it does not fulfill Eq. (2) and it does not represent the correct spatio-temporal dependence structure.

The next step is to simulate fields $U_k$ with $k = 1, \ldots, K$ which fulfill the homogeneous conditions, i.e. $U_k(x_j, t_i) = 0$. Further all fields $U_k$ need to share the same spatio-temporal dependence structure, again described by $\Gamma_c$. Such fields can be generated in a similar way as $W^*$ (see Hörning (2016) for details). The advantage of these fields $U_k$ is that they form a vector space (they are closed for multiplication and addition), thus:

$$W_\lambda = W^* + k(\lambda)(\lambda_1 U_1 + \ldots + \lambda_k U_k) \tag{8}$$

where $\lambda_k$-s denote arbitrary weights and $k(\lambda)$ denotes a scaling factor results in a field $W_\lambda$ which also fulfills the conditions prescribed in Eq. (1). The scaling factor is defined as:

$$k(\lambda) = \pm \sqrt{\frac{1 - \sum \alpha_l^2}{\sum \lambda_k^2}} \qquad (9)$$

It ensures that $W_\lambda$ exhibits the correct spatio-temporal dependence structure. Thus, transforming $W_\lambda$ back to $P$ using Eq. 5 will result in a precipitation field which has the correct spatio-temporal dependence structure, marginal distribution and honors the precipitation observations.

To also honor the observed runoff defined in Eq. 2 an optimization problem can be formulated:

$$\mathcal{O}(\lambda) = \sum_{i=1}^{I} (Q_t(G^{-1}(\Phi(W_\lambda))) - q_t)^2 \qquad (10)$$

which minimizes the difference between the modeled and observed runoff by optimizing the weights $\lambda_k$. As these weights are arbitrary they can be changed without violating any of the already fulfilled conditions, thus they can be optimized without any further constraints. If for a given set of fields and weights and after a certain number of iterations $N$ no suitable solution is found, the number $K$ of fields $U_k$ can be increased and the optimization is repeated. A suitable solution is found when the deviation between simulated and observed runoff is smaller than the criterion of acceptance $\varepsilon$ (here, $1 - NSE$ is used). If a suitable solution is found the whole procedure can be restarted using new random fields $V_l$. Thus multiple solutions can be obtained enabling uncertainty quantification of spatio-temporal rainfall fields.

## 3 Test of the methodology

### 3.1 Synthetic test site

To test the ability of the methodology a synthetic example was designed. The example consists of a synthetic catchment partly covered by rainfall. The synthetic catchment has a size of 211 km² with elevations range between 100 to 1100 m.a.s.l. and homogeneous landscape properties (Figure 2). A synthetic rainfall event of 6 hours duration with hourly time step and a maximum spatial extension of 118 km² on a regular grid of 1 km by 1 km cell size is used. Rainfall amounts above 20 mm/event covers an area of 25 km² with maximum rainfall of 36 mm/event and maximum intensity of 12 mm/h (see Figure 3 and Figure A1 in supplementary material). Based on this known spatio-temporal rainfall input pattern and RR-model parameterisation the catchment response at surface outlet was simulated and dedicated to be the known "observed" runoff $q_t$ (see Figure 6, blue graph).

Furthermore, ten different cells were selected from the spatio-temporal rainfall pattern to represent virtual monitoring stations of rainfall. They were chosen in a way that the centre of the event is not recorded. They are dedicated to be the known "observed" rainfall $P(x_j, t_i)$ at $J$ monitoring stations for $T$ time steps and provide the data basis for interpolation, conditional simulation, and inverse modeling of spatio-temporal rainfall pattern. Figure 4 shows their course in time. Note that virtual monitoring stations 2, 5, 9 and 10 measure 0 mm/h rainfall only. Based on these observations the fitted parameters for the

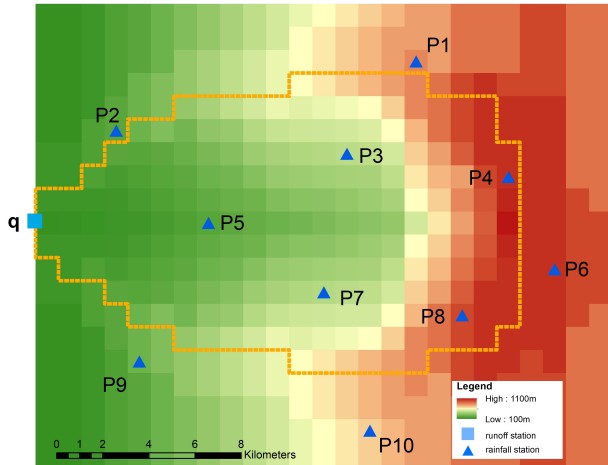

**Figure 2.** Topography, watershed and observation network of the synthetic catchment

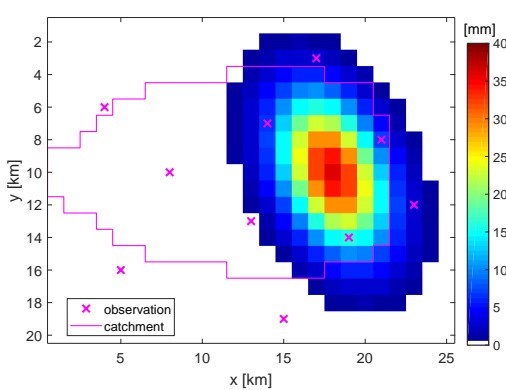

**Figure 3.** Rainfall amounts of the synthetic rainfall event. Virtual monitoring stations are marked by crosses.

marginal distribution (Eq.3) are: $p_0 = 0.36$ and $\lambda = 0.48$. The fitted copula for the dependency structure in space and time is a Gaussian copula with an exponential correlation function with a range of 2.5 km in space and a range of 1.5 h in time. In comparison, using the full synthetic dataset a range of 4.5 km in space and a range of 2.5 h in time are estimated.

## 3.2 Results and discussion

### 3.2.1 Common hydrologic modeling approach

At first, hourly rainfall data from virtual monitoring stations were used to interpolate the spatio-temporal rainfall pattern on a regular grid of 1 km by 1 km cell size by using the inverse distance method which is quite common in hydrologic modeling.

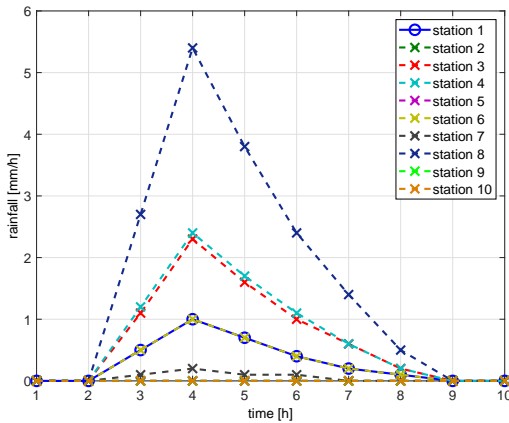

**Figure 4.** Time series of rainfall intensities at virtual monitoring stations

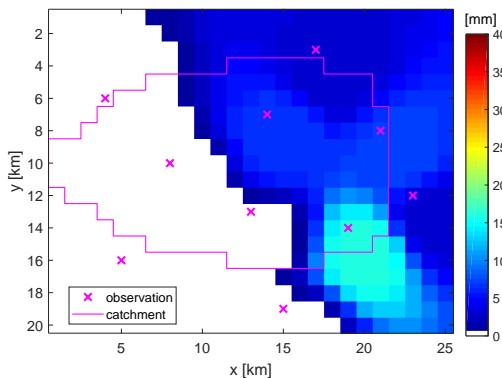

**Figure 5.** Interpolated rainfall amounts per event by using data of virtual monitoring stations

Afterwards, the response of the synthetic catchment was calculated by the RR-model. Figure 5 shows the interpolated pattern of the event based rainfall amounts as the sum over single time steps. The pattern looks quite smooth and has only minor similarities with the true pattern in Figure 3. Maximum of rainfall amount per event is equal to the maximum of the observation at virtual station number 8 with 16.2 mm/event. Therefore, the extension of a rainfall centre over 20 mm/event cannot be estimated. Due to low rainfall intensities, the simulated response of the RR-model shows a significant underestimation of the observed runoff with NSE value of -0.28 (see Figure 6, green graph).

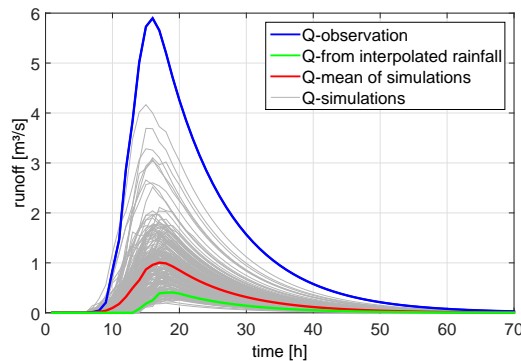

**Figure 6.** Runoff simulations based on simulated spatio-temporal rainfall pattern conditioned at rainfall point observations only (grey graphs) compared to its mean (red graph), runoff observation (blue graph), and simulation based on interpolated rainfall pattern (green graph)

### 3.2.2 Performance of conditional rainfall simulations

The random mixing approach was used to simulate 200 different spatio-temporal rainfall pattern conditioned on the virtual rainfall monitoring stations only. Resulting runoff simulations are displayed in Figure 6. They show a wide range of hydrographs with peak values between 0.19 m³/s to 4.17 m³/s and NSE values between -0.37 to 0.89. Compared to the runoff observation, the timing of peaks is acceptable, but the peak values are underestimated. Only four hydrographs have NSE values higher than 0.7. The corresponding spatial event based rainfall amounts for the top three runoff simulations regarding the NSE values ((a) 0.89 (b) 0.78 (c) 0.73) is shown in Figure 7. Their rainfall amounts ranging between 27.8 to 28.7 mm/event with a spatial extent of 9 to 11 km² of rainfall above 20 mm/event and a maximum intensity 10.5 to 15.1 mm/h. Compared to the observation (Figure 3), the spatial pattern look similar, at least regarding the spatial location of the event, and cover the maximum intensity. But the rainfall amounts per event as well as their spatial extent is too low. As a consequence, none of the simulated spatio-temporal rainfall fields conditioned at the virtual rainfall monitoring stations only are able to match the observed peak value in resulting runoff.

### 3.2.3 Inverse hydrologic modeling approach

The inverse modeling approach was used to simulate 107 different spatio-temporal rainfall pattern which are conditioned on the virtual rainfall and runoff monitoring stations, and runoff simulation results better than NSE values of 0.7. Afterwards a refinement have been carried out by selecting only those simulations with nearly identical runoff simulation results compared to observation. These simulations are characterized by NSE values larger than 0.995. Figure 8 shows the performance of the 20 selected realisations by grey graphs having only minor deviations during the flood peak range compared to the observation (blue graph). Associated rainfall patterns are displayed in Figure 9 for six selected realisations by their spatial rainfall amounts per event. Compared to the true spatial pattern (see Figure 3) none of them reproduce the true pattern exactly, but all of them locate the centre of the event in the same region as the true pattern. This shows, that by additional conditioning of spatio-

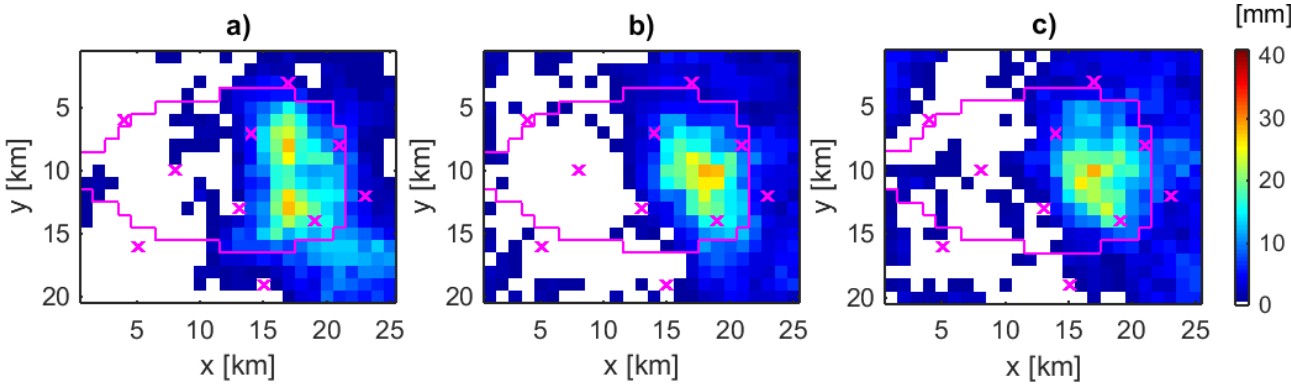

**Figure 7.** Event based rainfall pattern conditioned at rainfall point observations only for the top three runoff simulations in Figure 6

temporal rainfall pattern on runoff observation and consideration of catchment's drainage characteristic represented by the RR-model, the rainfall event can be localised and reconstructed in its spatial extent as well as in its course in time (see also Figure A1 in supplementary material). Most probably, if we would sample a large number of rainfall fields conditioned on rainfall observation only, we would find a realisation which matches the runoff observation too. Due to additional conditioning 5 on runoff we find these realisation faster.

However, the inference of a three dimensional input variable by using an integral output response results in a set or ensemble of different solutions. Rainfall amounts of the selected 20 realisations above 20 mm/event cover an area of 13 to 25 km² with maximum rainfall of 26.7 to 40.4 mm/event and maximum intensities of 10.7 to 17.1 mm/h. The event based areal precipitation of the catchment ranges between 98.2 % - 114.7 % of the observation (see Figure 3). Figure 9 presents spatial rainfall amounts 10 per event for: a) the realisation with the smallest area above 20 mm/event and smallest intensity, b) the realisation with the largest area above 20 mm/event c) the realisation with the highest intensity and rainfall amount per event, d) the realisation with the best NSE value in resulting runoff, e) and f) realisations with similar event statistics like the true spatio-temporal rainfall pattern. Compared to the observed pattern (see Figure 3), the different realisations match the spatial location as well as the shape of the observed pattern very good. However, the spatial patterns of the realisations are not such smooth and 15 symmetric like the constructed synthetic observation. Furthermore, the realisations show some scattered low rainfall amounts, which are not of importance for the hydrograph simulation, since they are addressed by the initial and constant rate losses of the RR-model.

Deriving an average rainfall pattern by calculation of the mean value per grid cell over all realisations of the ensemble for each time step, a smoother pattern is obtained, which looks more similar to the true one but has smaller rainfall intensities. 20 Using this mean ensemble pattern for calculating the runoff response, lead to an underestimation of the observed hydrograph as shown by the black hydrograph in Figure 8. Therefore, the ensemble mean of the hydrographs (red line in Figure 8) is a better representative for the sample than the mean ensemble rainfall pattern.

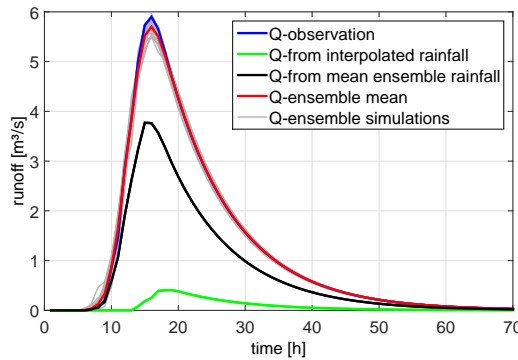

**Figure 8.** Comparison of hydrographs for the synthetic catchment shown by the observed runoff (blue) and rainfall-runoff simulation results based on: interpolated rainfall pattern (green), simulated ensemble of spatio-temporal rainfall pattern conditioned at rainfall and runoff observations (grey) and their mean value (red), as well as mean ensemble rainfall pattern (black)

In addition, data of the virtual monitoring stations (the observation) have been always reproduced and are equal for each rainfall simulation. This means, that each realisation reproduce the point observation of rainfall without any uncertainty. Only the grid points between the observation differs within the three-dimensional rainfall field and contain the stochasticity given by rainfall simulations conditioned on the observed values. In this context, the ensemble can be used as a partial descriptor of the
total uncertainty. It describes the remaining uncertainty of precipitation if all available data are exploited under the assumption of error-free measurements, reliable statistical rainfall models, and known hydrologic model parameters. .

## 4   Application for real world data

### 4.1   Arid catchment test site

The real world example is taken from the upper Wadi Bani Kharus in the northern part of the Sultanate of Oman. It is the
starting point for the present study and part of our multiyear research on hydrologic processes in this region. The headwater under consideration is the catchment of the stream flow gauging station of Al Awabi with an area of 257 km², located in the Hadjar mountain range with heights ranging from 600 m.a.s.l. to more than 2500 m.a.s.l. The geology of the area is dominated by the Hadjar group consisting of limestone and dolomite. The steep terrain consists of rocks mainly. Soils are negligible. However, larger units of alluvial depositions in the valleys are important for hydrologic processes which is addressed by spatial
differences in RR-model parameters. Vegetation is sparely and mostly cultivated in mountain oasis. Annual rainfall can reach more than 300 mm/year showing a huge variability between consecutive years. Analysis of measured runoff data over a period of 24 years shows that runoff occurred in average only on 18 days/year. Figure 10 displays the available monitoring network for sub daily data. Runoff is measured in 5 to 10 minutes temporal resolution. Rainfall measurements vary from 1 minute to 1 hour. Therefore, a temporal resolution of 1 hour was chosen for the event under investigation in this study. Figure 11 shows

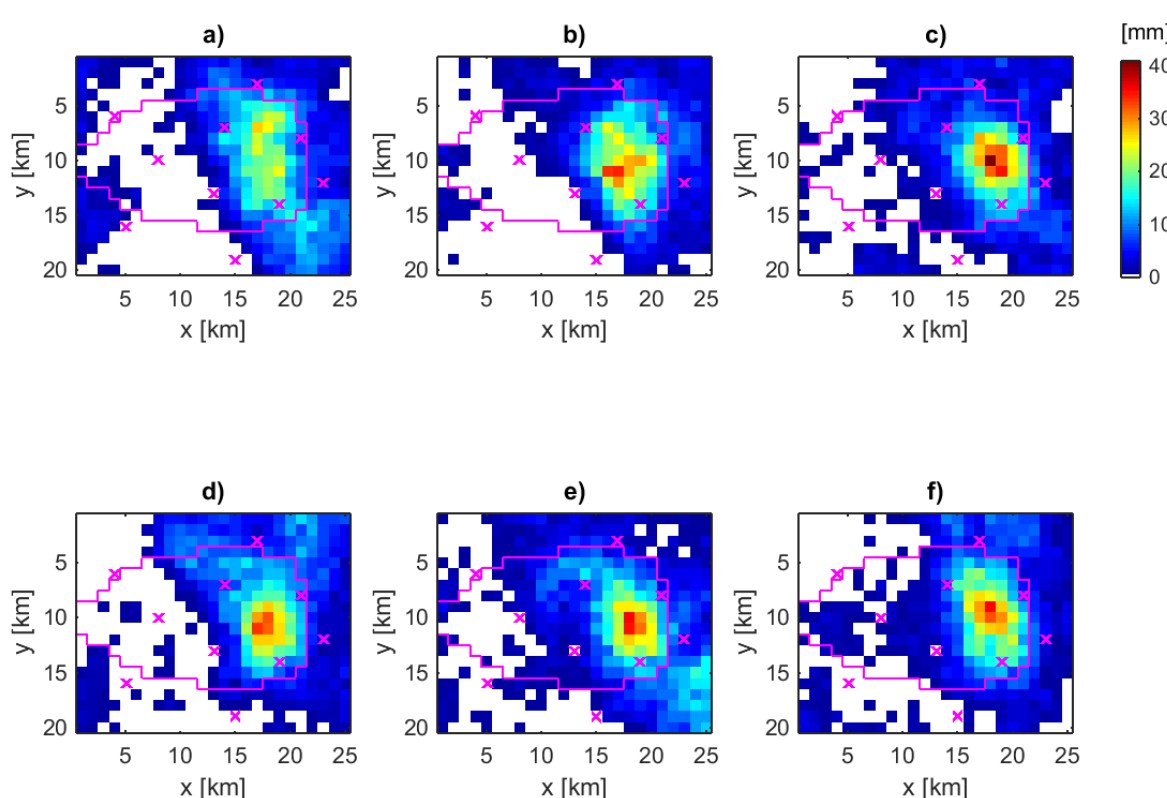

**Figure 9.** Selected realisations of spatial rainfall amounts per event with similar performance in resulting runoff obtained by the inverse modeling approch for simulating spatio-temporal rainfall pattern: a) realisation with the smallest area above 20 mm/event and smallest intensity, b) realisation with the largest area above 20 mm/event c) realisation with the highest intensity and rainfall amount per event, d) realisation with the best NSE value in resulting runoff, e) and f) realisations with similar event statistics like the true spatio-temporal rainfall pattern

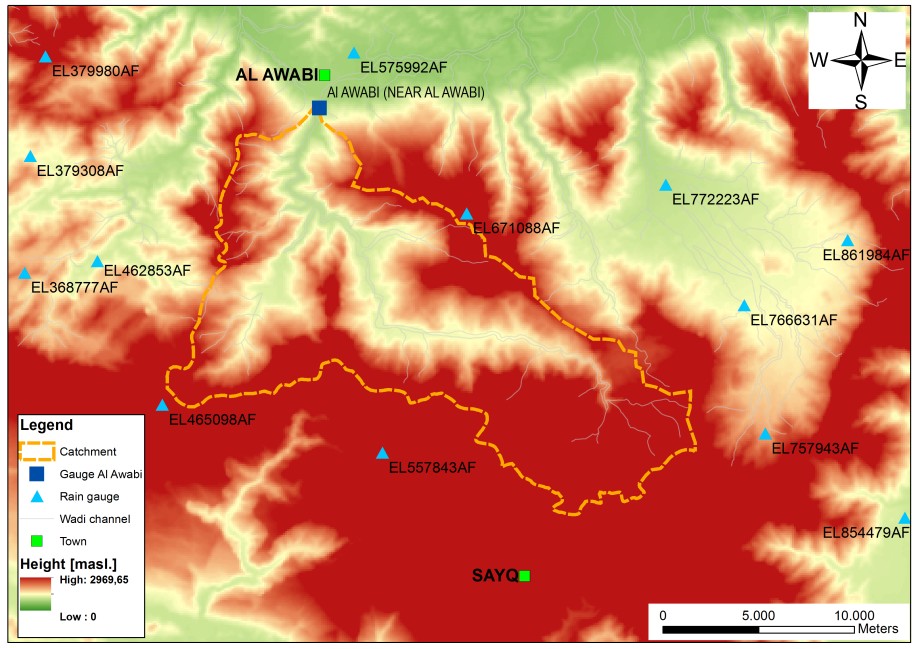

**Figure 10.** real world case study: catchment of gauge Al Awabi and sub daily monitoring network for runoff and rainfall

the measurements of the rainfall gauging stations and their altitudes for the rainstorm from 12 February 1999. Most of the rain was recorded on stations with lower altitudes located in the north-west and south-eastern part of the catchment. Rainfall interpolation was performed by inverse distance method, since there was no dependency of rainfall from altitude identifiable for this single heavy rainfall event. Parameters for the inverse modeling approach are: $p_0 = 0.17$ and $\lambda = 0.14$ for the marginal distribution (Eq.3). The fitted copula for the dependency structure in space and time is a Gaussian copula with an exponential correlation function with a range of 10 km in space and a range of 1 h in time.

## 4.2 Results and discussion

The real world data example was performed for the runoff event from 12 February 1999 with an effective rainfall duration of three hours. The simulated runoff for the interpolated rainfall pattern shows an underestimation of the peak discharge as well as a time shift of the peak arrival time compared to the observation (Figure 12). Applying the inverse approach by conditioning spatio-temporal rainfall pattern on rainfall and runoff observations, an ensemble of 58 different hydrographs is obtained after refinement having NSE values larger than 0.9. As shown in Figure 12, all of these hydrographs (grey graphs) represent the observation well and overcome the time shift. To explain this behaviour, differential maps are calculated which show the difference between the simulated and the interpolated rainfall pattern for each time step (Figure 13, see also Figure A2 in supplementary material for comparison of event based spatial rainfall amounts). It is easy to see that the inverse approach allows for a shift of the centre of the rainfall event from time step 1 to time step 2 and towards the catchment outlet. This

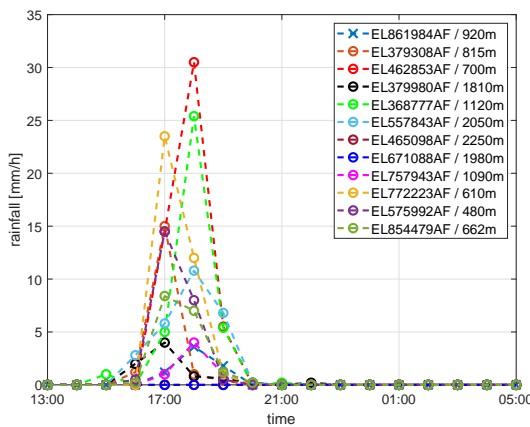

**Figure 11.** Rainfall amounts and altitudes of rainfall gauging stations from 12 February 1999

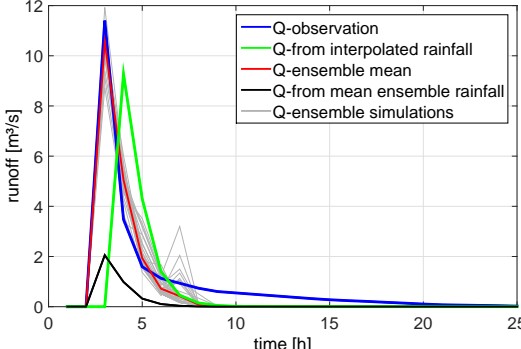

**Figure 12.** Comparison of hydrographs for the real world catchment shown by the observed runoff (blue) and rainfall-runoff simulation results based: on interpolated rainfall pattern (green), simulated ensemble of spatio-temporal rainfall pattern conditioned at rainfall and runoff observations (grey) and their mean value (red), as well as mean ensemble rainfall pattern (black)

results in a faster response of the catchment by its runoff compared to the interpolated rainfall pattern. In general, the obtained ensemble of spatio-temporal rainfall pattern is able to explain the observed runoff without discrepancy in rainfall measurements. Similar to the synthetic example, the ensemble mean hydrograph (Figure 12, red graph) is a better representative for the sample than the hydrograph based on the mean ensemble rainfall spatio-temporal pattern (black graph).

## 5  Summary and conclusions

An inverse hydrologic modeling approach for simulating spatio-temporal rainfall pattern is presented in this paper. The approach combines the conditional random field simulator Random Mixing and a spatial distributed RR-model in a joint Monte-

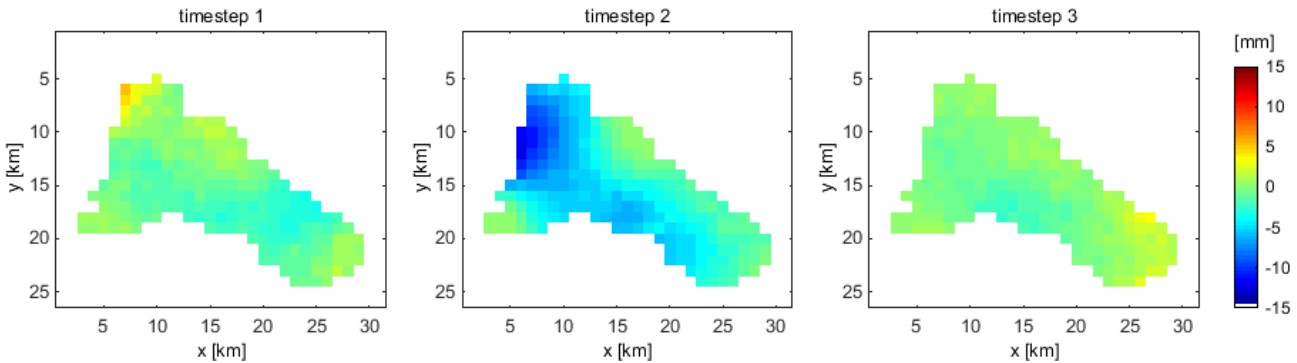

**Figure 13.** Differential maps of spatio-temporal rainfall pattern for three consecutive time steps (simulation – interpolation)

Carlo framework. It allows for obtaining reasonable spatio-temporal rainfall patterns conditioned on point rainfall and runoff observations. This has been demonstrated by a synthetic data example as well as a real world data example for single rainstorms and catchments which are covered by rainfall partly.

The proposed framework was compared to the methods of rainfall interpolation and conditional rainfall simulation. Recon-
5 struction of event based spatio-temporal rainfall pattern has been feasible by the inverse approach, if runoff observation and catchment's spatial drainage characteristic represented by the RR-model with spatial distributed travel times of overland flow are considered. As shown by the synthetic example, the rainfall pattern obtained by interpolation didn't match the observed rainfall field and runoff. If rain gauge observations don't portray the rain field adequately, a "good" interpolation result in least square sense is not a solution of the problem. This is the case in particular for small scale rainstorms with high spatio-
10 temporal rainfall variability and/or rainfall data scarcity due to insufficient monitoring network density. By rainfall simulations conditioned on rain gauge observation only, reasonable spatio-temporal rainfall fields are obtained, but with a wide spread in resulting runoff hydrographs. A large number of simulated rainfall fields is required to find those realisations which match the observed runoff, since the amount of possible conditioned rainfall fields is very much higher than the amount of rainfall fields matching point observation and runoff. By applying optimization, rainfall fields are conditioned on discharge too, and
15 appropriate candidates of spatio-temporal rainfall pattern can be identified more reliable, faster, and with reduced uncertainty.

The inference of a three dimensional input variable by using an integral output response results in a set of possible solutions in terms of spatio-temporal rainfall pattern. This ensemble is obtained by repetitive execution of the optimization step within the Monte-Carlo loop. It can be considered as a descriptor of the partial uncertainty resulting from spatio-temporal rainfall pattern estimates (under the assumption of error-free measurements, reliable statistical rainfall models, and known hydrologic
model parameters). Realisations of the ensemble vary in rainfall amounts, intensities, and spatial extend of the event, but they reproduce the point rainfall observation exactly and yield to similar runoff hydrographs. This allows for deeper insights in hydrologic model and catchment behavior and gives valuable information for the reanalysis of rainfall-runoff events, since rainstorm configurations leading to similar flood responses become visible. As shown in the example, operating with an en-

semble mean is less successful to match the runoff observation compared to an application of the whole ensemble due to smoothing effects.

The approach is also applicable under data scarce situation as demonstrated by a real world data example. Here, the flexibility of the approach becomes visible, since simulated rainfall pattern are also allow for overcoming a shift in timing of runoff. Therefore, the approach can be considered as a reanalysis tool for rainfall-runoff events especially in regions where runoff generation and formation are based on surface flow processes (Hortonian runoff), and catchments with wide ranges in arrival times at catchment outlet e.g. mountainous regions or distinct drainage structures e.g urban and peri-urban regions.

Nevertheless, further research and investigations are required. Examples presented in this paper are based on hourly resolution in time and 1 km² grid size in space. Especially for rainstorms in fast responding, small catchments finer resolutions in space and time are required. Here the limits of the approach in number of time steps and grid cells need to be explored. An other point is the required amount and quality of observation data as well as statistical model selection to obtain space-time rain fields. Both impact the simulation of rainfall amounts and of pattern by the derived spatial and temporal dependence structure. In these examples Gaussian copulas are used which might be not a good estimator for the spatial dependency in any case of heavy rainfall.

The proposed framework is a first step that only aims at reconstructing spatio-temporal rainfall pattern under the assumption of fixed hydrologic model structure and parameters. Certainly, hydrologic model uncertainty is of importance. But instead of changing the model to fit the observed discharge, we estimate rainfall fields which fit the model and the discharge by doing reverse hydrology. As such plausible rainfall fields can be identified, the corresponding model and the rainfall field is plausible. Thus, the framework can be applied to proof hypothesis about hydrologic model selection or to explain extraordinary rainfall-runoff events by using a well calibrated, spatial distributed hydrologic model for the catchment of interest. In this context, further research is dedicated to provide a common interface within the Monte-Carlo framework to exchange the hydrologic model and allow for broader use within the community. Also, further sources of uncertainties (e.g. model parameters, observations, ...) need to be considered to contribute for the solution of the hydrologic modeling uncertainty puzzle.

**Author contribution**

JG and AB conceived and designed the study. JG and SH performed the analysis and wrote the manuscript. AB contributed to the interpretation of the results, and commented on the manuscript.

**Data availability**

All data (except for confidential data) can be requested via email (jens.grundmann@tu-dresden.de).

**Competing interests**

The authors declare that they have no conflict of interest.

*Acknowledgements.* The corresponding author wishs to thank the Ministry of Regional Municipalities and Water Resources of the Sultanate of Oman for providing the data for the real case study and supporting the joint Omani-German IWRM-APPM initiative. Research for this paper was partly supported by the German Research Foundation (Deutsche Forschungsgemeinschaft, No.: 403207337, BA 1150/24-1 ) and partly by the Energi Simulation Program. Furthermore, we acknowledge support by the Open Access Publication Funds of the SLUB/TU Dresden.

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
