# Peer review of "Stochastic reconstruction of spatio-temporal rainfall pattern by inverse hydrologic modeling"

_Hydrology and Earth System Sciences, 2018_

## Referee Comment (RC1) · Anonymous Referee #1 · 9 Jul 2018

My comments are in the order I read the paper:

p2I3 - spelling mistake - "generall" p3I3 - some grammar issue - please check p6 - in my understanding, the approach is roughly along the following lines - first transformed empirical CDFs are ascertainedand used to create equivalent Normal observed rainfalls donated as w. line 11 says Gaussian copula is fitted to describe spatio-temporal dependence structure - a few lines of what this entails should be provided for completeness - in my reading this step seems independent of what is described next in the paper but if I am wrong this should be corrected. I am guessing the w\_j,i is not from this copula but from equation 4 for each site and time step. So you have L seguences and the aim is to fine alpha I such that there is some minimal deviation with the transformed normal rainfall at each location and timestep. So I guess the idea here is to keep generating fields until they match the observed rainfalls transformed to Normal. If that happens then you will have L=J and all the alpha's being equal to 1/L. And since there is spatial dependence, you would kind of expect L<J if this works fine. Am I correct? May be good to spell this out a bit more. P6l23 - what are homogenous conditions? I didnt understand what i meant by Uk(x\_j,t\_i)=0 - please clarify what this is and why is it needed. p7l2 - this is starting to become confusing now. Where did the covariance matrix come from? covariance of what? if W\* represents more or less the transformed observed precipitation field (from what I could gather), is this W lambda some randomised representation of that? If you are adding positive random values to this, arent you changing the probability distribution of W\_lambda from uniform to something shifted/tending to Gaussian? P7I7 - I presume this is a minimisation being performed which I think should attain a minimum value if the W\* is representing the observed precipitation field and the scaling weight k(lambda) equals zero. I am unclear about this approach - this is attempting to create the observed rainfall sequence instead of doing a stochastic generation as far as I can figure this out. P7I12 - the authors are saying multiple sequences are created by generating new random fields VI and enabling something called uncertainty guantification - please explain what this means. I am very curious how different the sequences end up being - and when they are really different, whether their probability distributions are consistent with the observed series that was used. Also - am I correct in stating that the timing of these sequences will be fairly similar to the observed sequence - hence the final sequences will be representing uncertainty about each observed value more than representing a stochastic system that is generating equally plausible sequences (a bit like a weather generator does conditional to exogenous inputs, compared to a stochastic generator where no two sequences have any exogenous binding variable). p7l15 - Am I correct in interpreting that the ranfall is generated known the marginal distribution at each pixel of the 118km2 catchment? Or is it based on the 6 hours of rainfall at the 10 monitoring

stations alone? If it is the latter, assumptions must have been made to spatially interpolate/extrapolate the rainfall to other pixels. Please state these. If it is the former, this is a limitation I believe as you need to be sure about the spatio-temporal structure of your storm to help refine it further using the flows. P7I26 - some mention of the number of time steps in the observed record for rainfall and flows should be provided - there is a mention of 6 hours but I wasnt sure if that is the time step of the duration. P9fig6 - I see all hydrographs are having roughly the same timing of the peak. So what I suspected about the time sequences of the rainfall is most likely correct. The differences across the storms would not be significant in terms of the spatial or the temporal pattern uncertainty that exists in real cases. I think this could be a limitation if the approach were being pitched as a stochastic generator - but could form an interesting way to generate alternate realisations of a storm sampled at specified point locations alone. And the need for having an accurate hydrologic model is a big limitation too as the uncertainty that arises from this can be significant.

On the whole, I am unclear how I would use an approach such as this for my modelling application. I will need to have a fairly good idea of the spatio-temporal nature of the storm system to put this into use - along with having point rainfalls and modelled flow time series to help ascertain which sequences are good. I think the authors need to add more examples of this in their revision to establish a clear scenario how users will put their method into use. And some details of the tolerences etc that are used to make this stochastic should be added as I think they are not stated in the paper very clearly. Some indication of how this might perform over long storms/large catchments/very few point locations etc will really help readers.

---

## Referee Comment (RC2) · Anonymous Referee #2 · 16 Jul 2018

The paper 'Stochastic reconstruction of spatio-temporal rainfall pattern by inverse hydrological modelling' by Grundmann J., Hörning, S. and Bárdossy, A. proposes a method to estimate high resolution space-time rain fields from sparse rain gauges observations complemented by streamflow measurements. I find the idea of incorporating streamflow measurements and inverse hydrological modelling to reconstruct rain fields very interesting. And to my knowledge it is the first time that it is proposed to apply this idea to the reconstruction of high resolution space-time rain fields. In that respect I find this paper original. In addition the topic is relevant for the readers of HESS.

However, I feel that in the present version of the manuscript, the authors do not provide

enough information (and of sufficient quality) to be able to assess the proposed framework. In addition I have the impression that even if interesting, the proposed approach cannot reach all the targets stated by the authors.

To sum up, I have the feeling that this paper addresses an interesting idea, but the current version is very preliminary (too much in my opinion) and does not allow to capitalize on the framework developed by the authors.

I start by listing the points I would need to know in order to fully understand and assess the proposed method. After that, I will detail some concerns I have about the method itself. Afterwards I finish my review by few minor comments.

Possible improvements to better explain the method:

* First of all, the written English must be improved. The present version of the manuscript is full of errors that shocked me even though I am not a native English speaker. At a minimum, a spell checker must be used. When I applied mine to the present manuscript I obtained dozens of errors and typos... In addition, some sentences are grammatically incorrect or difficult to understand. For instance: p1L20-24, p3L3-4, p11L5-8.

* Regarding the introduction and the context of this study, I acknowledge that the application of inverse modelling to the reconstruction of space-time rain fields is new. However the idea of inverse hydrology in general (i.e. without space-time application) has already been proposed by several authors, as well as the idea of using streamflow data to improve rainfall input estimation. Unfortunately, none of these works are mentioned in the introduction. I find it quite unfair. I really would like to see more background about previous studies addressing similar ideas in order to better contextualize the present study. I can suggest for instance the following papers (I didn't participate to these works):

- Kirchner J.W. (2009): Catchments as simple dynamical systems: Catchment characterization, rainfall-runoff modeling, and doing hydrology backward, Water Resources Research, 45, W02429, doi:10.1029/2008WR006912.

- Kretzschmar A. et al (2014): Reversing hydrology: Estimation of sub-hourly rainfall time-series from streamflow, Environmental Modelling and software, 60, 291-301.

-Del Giudice, D. et al (2016): Describing the catchment-averaged precipitation as a stochastic process improves parameter and input estimation, Water Resources Research, 52, 3162-3186, doi:10.1002/2015WR017871.

* Regarding the description of the rainfall-runoff model, very few information is provided. What is specified is basically that it is a distributed model, no more. For instance I don't know the name of the model, there is no reference about this model, and no equation to explain how it works. However I am sure that the hydrological model used for the inversion of the streamflow to reconstruct rainfall has a significant impact on the final result. By the way, the impact of the choice of the hydrological model (e.g. distributed vs semi-lumped) should be discussed somewhere in the paper.

* Regarding the description of the Random Mixing approach, I really lack information about the underlying statistical model and the inference of its parameters. To be honest I had to read the paper of Haese et al (2017) to be able to understand the application of Random Mixing to rainfall modelling. Therefore I think that not enough efforts have been made to explain the Random Mixing method in the present paper. In particular I would be interested to know:

- Which spatio-temporal copula is used? Does it need to be a valid covariance function (or is it irrelevant in the context of copulas)?

- How are the parameters of the model (i.e. the marginal transform function and the copulas) inferred in practice? In particular how do you deal with dry measurements (i.e. rain intensity=0) in the inference process? (I think it is important here since rain intermittency can be significant in semi-arid and arid regions). Ok there is a reference

to Li (2010), but more information within this paper would be a plus for the reader.

- Which simulation method is used in practice to generate the unconditional simulations? You cite several methods but I would like to know the one you are actually using.

* Regarding the synthetic case study, it is not clear to me if the parameters of the statistical rainfall model used in the random mixing are inferred from the synthetic data. I suppose that it is the case, but it should be clearly mentioned. If it is the case, it would be interesting to show the results of the fitting procedure. For instance: which copula (with which parameters) has been fitted? And also which marginal distribution? And how do the estimated values of the model parameters compare with the true ones (in this case you know the true values because it is a synthetic case)? In fact I suspect that the inferred statistical rainfall model cannot capture properly the true statistics of rainfall because the center of the rain cell is not observed. This can explain why conditional simulations (without streamflow constraints) cannot reproduce the observed hydrograph. I will come back to this point in my concerns about the method.

* Regarding the real world application. I would have been more convinced if you have shown an example with cross-validation. For instance the reconstruction of space-time rainfall for a well instrumented catchment (with many rain gauges). In this case you can select some stations for the inference of the mixing model parameters and the estimation of rain fields, and keep other stations to cross-validate the rain estimations. In addition, in the real world application, the altitude of the catchment ranges from 600m to 2500m; in this case one can expect some non-stationarities in rainfall statistics. Could you please discuss a bit this potential issue?

Concerns about the method itself:

* In the proposed method, the parameters of the hydrological model are supposed to be known and fixed. But at the same time the goal is to infer high resolution space-time rain fields to... improve hydrological modelling. This seems a bit circular reasoning. I see two options to break the circle:

[Figure]

- Either clearly acknowledge that the proposed framework is a first step that only aims at reconstructing space-time rain fields from rain gauge and streamflow measurements. Basically a proof of concept with strong assumptions (incl. known hydrological model), that will be relaxed only in future work. And in this case do not claim that the goal is to improve hydrological modelling, but just to show that doing reverse hydrology to reconstruct space-time rain fields is somehow feasible. In my opinion this is already a very nice contribution.

- Or improve the proposed framework to jointly reconstruct space-time rain fields and calibrate the hydrological model. This can be seen as the extension of the work of Del Giudice et al (2016) (see ref above) to the case of space-time rain fields. But I suspect that this will require a lot of developments... and I am not sure it will work in many configurations... But if it works it would be an even nicer contribution.

* In the synthetic case study, runoff simulations based on simulated rain fields conditioned to point observation only do not encompass the ones based on conditioning to rain gauge observations and streamflow observations. This is clearly visible by comparing figures 6 and 8. I am very surprised about it. Indeed, in my understanding, the second case (adding conditioning to streamflow) should just add constraints to the first case. Therefore it should only select the rain fields obtained by simulation conditional to rain gauge only that are compatible with streamflow observations. But it is clearly not the case here... Therefore either I am missing something, and in this case I believe the reasons why this result arises must be explained in much more details by the authors; or there is some issue. One possible reason I could suggest (but I am not sure) is that the actual rain field that generates the observed hydrograph is kind of an extreme of the multivariate statistical distribution that underlies the random mixing model (after fitting model parameters). And therefore this extreme is not sampled by the 200 realizations performed in Figure 6.

* Regarding the assessment of uncertainty, I would be more cautious before stating "This ensemble can be used to describe the uncertainty in estimating spatio-temporal

rainfall patterns" (p11, L9). In my opinion, the ensemble of realizations that is obtained is only a very partial descriptor of the total uncertainty. Indeed, in the proposed framework, both the statistical rainfall model and the runoff generation model have fixed parameters. Therefore the uncertainty originating from these two components is neglected. In the end, only the uncertainty related to the scarcity of the rain gauge measurement network is accounted for. I think this should be more clearly explained to the reader.

Minor comments:

-P2L22: In my knowledge, the turning band method is linked to the theory of random fields rather than to point processes.

-P2L34: "with respect to the outlined problem in the second paragraph above" -> not clear what you are referring to.

-Modeling vs modelling: you have to choose one spelling.

-Eq 2: why qn and not q(t)?

-P4L20: It would be more clear if you say that P(x,t) is precipitation instead of rainfall. Or maybe call the variable R?

-Eq 3: Don't mix P and p.

-Eq 4: Don't mix W and w.

-Eq 6: You should mention that conditioning is made at this step.

-Figure 3, 5, 7, 9: Please add units to the X and Y axes as well as to the color bar. You could also add the limits of the watershed.

-Real case study: you should show the observation dataset.

---

## Author Comment (AC1) · 24 Sep 2018

article [utf8]inputenc geometry verbose,tmargin=3cm,bmargin=3cm,lmargin=3cm color textcomp

**Authors final comments to referee 1**

September 24, 2018

Journal: HESS

Title: Stochastic reconstruction of spatio-temporal rainfall pattern by inverse hydrologic modelling

Author(s): Jens Grundmann, Sebastian Hörning, András Bárdossy

MS No.: hess-2018-350

We would like to thank the referee for his/her time to review the manuscript. Our reply is organized as follows: (1) comments from Referee are in black color, (2) author's response is marked in blue color and placed within referee comments whenever it's needed, (3) author's changes in manuscript based on comments of both referees are summarized at the end of this document.

REFEREE 1:

My comments are in the order I read the paper:

p2l3 - spelling mistake - 'generall' Thanks, the spelling mistakes will be fixed in the revised manuscript.

p3l3 - some grammar issue - please check

Grammar will be improved throughout the whole paper. In general spelling and grammar will be double checked by a native speaker prior to resubmission.

p6 - in my understanding, the approach is roughly along the following lines - first transformed empirical CDFs are ascertained and used to create equivalent Normal observed rainfalls donated as w.

We are not quite sure what you mean by transformed empirical cdfs are ascertained. What is done is that a cdf (and in general this cdf can be any type of cdf, i.e. parametric or non-parametric or a combination of both) is fitted to the observed precipitation values. The distribution used in this work is described in Eq. 3. It is a combination of a discrete probability for zero precipitation values and an exponential distribution for values greater than zero. Thus the parameters that need to be estimated are  $p_0$  (the discrete probability of zero precipitation) and  $\lambda$  (the parameter of the exponential distribution).Subsequently, using this fitted cdf the observed precipitation values are transformed to standard normal values according to Eq. 4.

line 11 says Gaussian copula is fitted to describe spatio-temporal dependence structure - a few lines of what this entails should be provided for completeness. We are going to add more information on copulas in general, the Gaussian copula and the fitting process to the revised manuscript.

- in my reading this step seems independent of what is described next in the paper but

СЗ

if I am wrong this should be corrected. I am guessing the  $w_{j,i}$  is not from this copula but from equation 4 for each site and time step. So you have L sequences and the aim is to fine  $\alpha_l$  such that there is some minimal deviation with the transformed normal rainfall at each location and time step. So I guess the idea here is to keep generating fields until they match the observed rainfalls transformed to Normal. If that happens then you will have L = J and all the alpha's being equal to 1/L. And since there is spatial dependence, you would kind of expect L < J if this works fine. Am I correct? May be good to spell this out a bit more.

Yes, to some extent but there also seems to be some misunderstanding. The  $w_{j,i}$ -s (which represent the transformed precipitation observations at locations x and time steps t) are derived by Eq. 4 (see your third comment). The random fields  $V_l$  (which are independent standard normal spatial random fields) are simulated such that they all have the same spatial structure which is described by this Gaussian copula (this information is missing in the paper). Eq. 6 says that we want to find a linear combination of these independent standard normal random fields  $V_l$  such that this linear combination results in the values  $w_{j,i}$  at locations x and time steps t. Thus Eq. 6 describes a linear equation system with the weights  $\alpha_l$  being the unknowns (the values  $V_l(x_j, t_i)$  are known). This equation system can be solved for  $L \ge J$ , the bigger L is the smaller the  $\sum \alpha_l^2$  sum gets - if the solution is calculated using SVD.

P6l23 - what are homogenous conditions? I didn't understand what is meant by  $U_k(x_i, t_i) = 0$  - please clarify what this is and why is it needed.

The homogeneous conditions are  $U_k(x_j, t_i) = 0$  (a system of linear equations with the right hand side being all zeros is called a homogeneous equation system). This means that now we want to find a linear combination of random fields which fulfills  $U_k(x_j, t_i) = 0$ , i.e. a linear combination that results in zeros at locations xand time steps t. This is done the same way as constructing the field  $W^*$ , i.e. by setting up an equation system using independent standard normal random fields  $\sum_{k=1}^{K} \beta_k V_k(x_j, t_i) = 0$  (we didn't put this equation in the paper as it's basically the same as Eq. 6 with the right-hand side being zero). The explanation why this is needed is actually given in the following sentences. Line 24: 'The advantage of these fields  $U_k$ is that they form a vector space (they are closed for multiplication and addition)...' This means when adding such a field  $U_k$  (or k of them) to  $W^*$ , the resulting field  $W_\lambda$  will exhibit the correct values  $w_{j,i}$  at locations x and times t because the zeros in the field  $U_k$  do not affect these values. However, the rest of the field is affected (as fields  $U_k$  are conditional random fields) which enables modifying the final field  $W_\lambda$  without changing the conditioning values. By changing the arbitrary weights  $\lambda$  one can modify the field such that it represents the observed runoff (and therefore the procedure needs to be coupled with the rainfall runoff model) to a certain degree.

p7l2 - this is starting to become confusing now. Where did the covariance matrix come from? covariance of what?

This goes back to the missing information that the fields  $V_l$  all have the same spatial structure which is described by the fitted Gaussian copula. The covariance we are referring to is the spatio-temporal covariance of the observations to which we have fitted the Gaussian copula. We are going to change the wording (as it isn't consistent) and add more information to the revised manuscript. As the field that fulfills the homogeneous conditions can be combined using arbitrary weights  $\lambda$  the scaling factor  $k(\lambda)$  can be used to scale the final field such that the resulting field exhibits the spatio-temporal correlation/covariance of that copula. This is the case when the  $L^2$  norm of the weights of the linear combination is equal to 1. As  $\sum \alpha_l^2 << 1$  the weights  $\lambda$  need to be scaled (using the scaling factor  $k(\lambda)$ ) such that  $\sum \alpha_l^2 + \sum \lambda_k^2 = 1$ . It's also worth mentioning that in this case the covariance is equal to the correlation as we are working in standard normal space (mean is zero and unit variance).

if W\* represents more or less the transformed observed precipitation field (from what I could gather), is this  $W_{\lambda}$  some randomised representation of that? If you are adding

positive random values to this, arent you changing the probability distribution of  $W_{\lambda}$  from uniform to something shifted/tending to Gaussian?

 $W^*$  is already a random representation of a precipitation field that is conditioned on the available point observations. Due to the additional constraint  $\sum \alpha_l^2 << 1$  it however is a very smooth field (like an interpolated field), i.e. it does not represent the observed spatial variability of the precipitation. By adding the fields  $U_1, \ldots, U_k$  to  $W^*$  one can easily scale these fields to have the correct spatial dependence without modifying the observed values at the observation locations (because the weights  $\lambda$ are arbitrary as the fields  $U_1, \ldots, U_k$  fulfill the homogeneous conditions) such that the final field  $W_\lambda$  exhibits the correct spatial variability. Further, each realization of  $W_\lambda$ (e.g. by taking different  $\lambda$ ) is a conditional random field, i.e. a possible representation of the precipitation field. We are not adding positive random values and we are also not working with a uniform distribution but with a standard normal distribution (Eq. 4). This standard normal distribution doesn't change due to the linear combinations (zero mean will always remain zero mean in this case and the unit variance is ensured due to the scaling of the weights  $\lambda$ . Precipitation fields are obtained via back-transformation of these fields.

P7I7 - I presume this is a minimisation being performed which I think should attain a minimum value if the W\* is representing the observed precipitation field and the scaling weight  $k(\lambda)$  equals zero. I am unclear about this approach - this is attempting to create the observed rainfall sequence instead of doing a stochastic generation as far as I can figure this out.

There seems to be another misunderstanding. The described approach is a stochastic procedure as all fields used are random fields. We do not try to create the observed rainfall sequence except that we want to represent point observations as well as the observed runoff. Thus we are working with conditional random fields. As described above the weights  $\lambda$  are arbitrary if we only intend to reproduce the observed precipitation at the observation locations and the spatial variability. From these  $\lambda$  weights we

identify those which also reproduce the discharge. Thus the optimization described here is a function of these  $\lambda$  (and because it is an unconstrained optimization it is straight forward). In simple words, the field  $W_{\lambda}$  (which is already conditioned on precipitation observations) is modified such that the resulting simulated runoff (by the RR-model) is close to the observed runoff.

P7I12 - the authors are saying multiple sequences are created by generating new random fields  $V_l$  and enabling something called uncertainty quantification - please explain what this means. I am very curious how different the sequences end up being - and when they are really different, whether their probability distributions are consistent with the observed series that was used. Also - am I correct in stating that the timing of these sequences will be fairly similar to the observed sequence - hence the final sequences will be representing uncertainty about each observed value more than representing a stochastic system that is generating equally plausible sequences (a bit like a weather generator does conditional to exogenous inputs, compared to a stochastic generator where no two sequences have any exogenous binding variable). Yes this sentence should explained a bit more. It is mentioned in P3L7 that "... Our goal here is an event based reconstruction of possible realizations of spatio-temporal rainfall patterns which are conform with the measured point rainfall data and catchment runoff response at best. For that we are looking for potential candidates of three-dimensional (space-time) rainfall fields for sub daily time steps and spatial resolution of 1km2 ... ". This means that each candidate (or sequence) reproduce the point observation of rainfall without any uncertainty (or deviation). Only the grid points between the observation differs within the 3D rainfall field and contain the stochasticity given by simulations conditioned on the observed values.

p7I15 - Am I correct in interpreting that the rainfall is generated known the marginal distribution at each pixel of the 118km2 catchment? Or is it based on the 6 hours of

rainfall at the 10 monitoring stations alone? If it is the latter, assumptions must have been made to spatially interpolate/extrapolate the rainfall to other pixels. Please state these. If it is the former, this is a limitation I believe as you need to be sure about the spatio-temporal structure of your storm to help refine it further using the flows.

We are not quite sure what you mean with assumptions must have been made? Do you mean assumption must have been made to generate the synthetic reality? Or assumptions must have been made to generate possible realizations based on the 6 hours of rainfall at the 10 monitoring stations? If it is the latter then the assumptions that we made are that we can fit a marginal distribution and a spatial copula to these observations. Therefore only the values at the rainfall monitoring stations are used for the fitting etc. in order to make the synthetic test case a realistic scenario. But since this a synthetic test case all values at each pixel are known which enables comparison of the simulated results with the synthetic reality. We assume that the 6h precipitation distribution for the whole area is the same as the precipitation distribution derived from the observations (corresponding to the observation).

P7I26 - some mention of the number of time steps in the observed record for rainfall and flows should be provided - there is a mention of 6 hours but I wasnt sure if that is the time step of the duration.

It is already mentioned in the manuscript nine lines above (P7L17: "A synthetic rainfall event of 6 hours duration with hourly time step ...")

P9fig6 - I see all hydrographs are having roughly the same timing of the peak. So what I suspected about the time sequences of the rainfall is most likely correct. The differences across the storms would not be significant in terms of the spatial or the temporal pattern uncertainty that exists in real cases. I think this could be a limitation if the approach were being pitched as a stochastic generator - but could form an interesting way to generate alternate realisations of a storm sampled at specified point

locations alone.

The goal of the work is an stochastic "reconstruction" of spatio-temporal rainfall pattern ... (see title) which seams to be similar to what you called "to generate alternate realisations of a storm sampled at specified point locations alone". We are not interested in exploring overall spatio-temporal pattern uncertainty (e.g. by performing unconditional stochastic simulations and considering measurement uncertainty too) since this was already done in research and has no benefit for the focus of this paper. Fig 6 shows the results of 200 simulated spatio-temporal rainfall pattern conditioned at rainfall point observations only, but containing the spatial uncertainty for the unobserved points. The hydrographs have to look a bit similar since all simulations used the same rainfall values at observation points transformed into runoff by the same hydrologic model (representing the hydrologic properties of the catchment).

And the need for having an accurate hydrologic model is a big limitation too as the uncertainty that arises from this can be significant.

Of course hydrologic model uncertainity plays an important role, but instead of changing the model to fit the observed discharge we estimate rainfall fields which fit the model and the discharge. As such plausible rainfall fields can be identified, the corresponding model and the rainfall field is plausible.

On the whole, I am unclear how I would use an approach such as this for my modelling application. I will need to have a fairly good idea of the spatio-temporal nature of the storm system to put this into use - along with having point rainfalls and modelled flow time series to help ascertain which sequences are good.

Some hints are given in the the summary section. (see P15L13 "... a reanalysis tool for rainfall-runoff events especially in regions where runoff generation and formation based on surface flow processes and catchments with wide ranges in arrival times at catchment outlet ..." or P15L22 "... where modelers are interested to explain the

extraordinary rainfall-runoff events ...". ) However, this section will be discussed more detailed in the revised manuscript.

I think the authors need to add more examples of this in their revision to establish a clear scenario how users will put their method into use. And some details of the tolerences etc that are used to make this stochastic should be added as I think they are not stated in the paper very clearly. Some indication of how this might perform over long storms/large catchments/very few point locations etc will really help readers We are not sure, what you mean by "tolerences". In general, the manuscript aims to present a new method and to show that it can deal with real world data. However, it is basic research and we are also very curious to explore the method further (see outlook P15L16). But this requires further developments (e.g. common interfaces for data, models, other types of copulas) which are not manageable within the next months. Among others we intended to show that models may be good even without any strong modification if we take the uncertainty of the precipitation into account. Thus models may help to improve precipitation estimation and one could consider model calibration under consideration of precipitation uncertainty.

**author's changes in manuscript**

Some hints regarding author's changes in manuscript have been already given in the comments section. Here, a summary of author's changes in manuscript based on comments of both referees is given.

 chapter 1 "Motivation": adding and discussion of literature regarding inverse hydrologic modeling, improved reasoning

- · chapter 2.2 "Rainfall runoff model": description will be improved
- chapter 2.3 "Random Mixing for inverse hydrologic modeling": description will be improved
- chapter 3.2 "Results and discussion": discussion of the synthetic example will be enhanced and performed more precise.
- chapter 4.1 "Arid catchment test site": figure of rain gauge measurements and additional information will be added
- chapter 4.2 "Results and discussion" discussion of the real case study will be enhanced and performed more precise
- chapter 5 "Summary and conclusion" reasoning will be enhanced and performed more precise
- revision of Figures 3,5,7,9
- · all typos will be fixed and grammar will be improved

---

## Author Comment (AC2) · 24 Sep 2018

article [utf8]inputenc geometry verbose,tmargin=3cm,bmargin=3cm,lmargin=3cm color float graphicx

**HESSD**

**Authors final comments to referee 2**

September 24, 2018

Journal: HESS

Title: Stochastic reconstruction of spatio-temporal rainfall pattern by inverse hydrologic modelling

Author(s): Jens Grundmann, Sebastian Hörning, András Bárdossy

MS No.: hess-2018-350

We would like to thank the referee for his/her time to review the manuscript. Our reply is organized as follows: (1) comments from Referee are in black color, (2) author's response is marked in blue color and placed within referee comments whenever it's needed, (3) author's changes in manuscript based on comments of both referees are summarized at the end of this document.

**REFEREE 2:**

The paper 'Stochastic reconstruction of spatio-temporal rainfall pattern by inverse hydrological modelling' by Grundmann J., Hörning, S. and Bárdossy, A. proposes a
method to estimate high resolution space-time rain fields from sparse rain gauges observations complemented by streamflow measurements. I find the idea of incorporating streamflow measurements and inverse hydrological modelling to reconstruct rain fields very interesting. And to my knowledge it is the first time that it is proposed to apply this idea to the reconstruction of high resolution space-time rain fields. In that respect I find this paper original. In addition the topic is relevant for the readers of HESS.

However, I feel that in the present version of the manuscript, the authors do not provide enough information (and of sufficient quality) to be able to assess the proposed framework. In addition I have the impression that even if interesting, the proposed approach cannot reach all the targets stated by the authors.

To sum up, I have the feeling that this paper addresses an interesting idea, but the current version is very preliminary (too much in my opinion) and does not allow to capitalize on the framework developed by the authors. I start by listing the points I would need to know in order to fully understand and assess the proposed method. After that, I will detail some concerns I have about the method itself. Afterwards I finish my review by few minor comments.

Possible improvements to better explain the method:

First of all, the written English must be improved. The present version of the manuscript is full of errors that shocked me even though I am not a native English speaker. At a minimum, a spell checker must be used. When I applied mine to the present manuscript I obtained dozens of errors and typos... In addition, some sentences are grammatically incorrect or difficult to understand. For instance: p1L20-24, p3L3-4, p11L5-8.

We are going to fix all typos and improve the grammar in the revised manuscript. The revised manuscript will be double checked by a native speaker.
Regarding the introduction and the context of this study, I acknowledge that the application of inverse modelling to the reconstruction of space-time rain fields is new. However the idea of inverse hydrology in general (i.e. without space-time application) has already been proposed by several authors, as well as the idea of using streamflow data to improve rainfall input estimation. Unfortunately, none of these works are mentioned in the introduction. I find it quite unfair. I really would like to see more background about previous studies addressing similar ideas in order to better contextualize the present study. I can suggest for instance the following papers (I didn't participate to these works):

- Kirchner J.W. (2009): Catchments as simple dynamical systems: Catchment characterization, rainfall-runoff modeling, and doing hydrology backward, Water Resources Research, 45, W02429, doi:10.1029/2008WR006912.

- Kretzschmar A. et al (2014): Reversing hydrology: Estimation of sub-hourly rainfall time-series from streamflow, Environmental Modelling and software, 60, 291-301. -Del Giudice, D. et al (2016): Describing the catchment-averaged precipitation as a stochastic process improves parameter and input estimation, Water Resources Research, 52, 3162-3186, doi:10.1002/2015WR017871.

Thank you very much for these references. We will improve the introduction and broaden the literature review and discussion.

Regarding the description of the rainfall-runoff model, very few information is provided. What is specified is basically that it is a distributed model, no more. For instance I don't know the name of the model, there is no reference about this model, and no equation to explain how it works. However I am sure that the hydrological model used for the inversion of the streamflow to reconstruct rainfall has a significant impact on the final result. By the way, the impact of the choice of the hydrological model (e.g. distributed vs semi-lumped) should be discussed somewhere in the paper.

**HESSD**
We will add additional information in the revised manuscript. Up to now, the model has no name. It uses only simple approaches known from hydrologic textbooks for the simulation of single events (no long-term water balance). It focuses on hortonion runoff and considers spatial distributed travel times for surface runoff. You are right, the choice of the model has impact on its results. We will enhance the discussion of this issue in the last section.

Regarding the description of the Random Mixing approach, I really lack information about the underlying statistical model and the inference of its parameters. To be honest I had to read the paper of Haese et al (2017) to be able to understand the application of Random Mixing to rainfall modelling. Therefore I think that not enough efforts have been made to explain the Random Mixing method in the present paper. In particular I would be interested to know:

Which spatio-temporal copula is used? Does it need to be a valid covariance function (or is it irrelevant in the context of copulas)?
We have used a Gaussian copula. And yes it needs to be a valid covariance function.
We are going to add more information on copulas in general and the Gaussian copula in the revised manuscript.

- How are the parameters of the model (i.e. the marginal transform function and the copulas) inferred in practice? In particular how do you deal with dry measurements (i.e. rain intensity=0) in the inference process? (I think it is important here since rain intermittency can be significant in semi-arid and arid regions). Ok there is a reference to Li (2010), but more information within this paper would be a plus for the reader. We are going to add a bit more information (and more references) on the inference process however we do not want to go into great detail as this is not the main focus of this work.

**HESSD**
- Which simulation method is used in practice to generate the unconditional simulations? You cite several methods but I would like to know the one you are actually using.

We have actually used the spectral representation method: Shinozuka, M., and G. Deodatis (1996), Simulation of multi-dimensional gaussian stochastic fields by spectral representation, Appl Mech Rev, 49(1). It is not in the references list yet so we are going to add it and mention it in the revised manuscript.

Regarding the synthetic case study, it is not clear to me if the parameters of the statistical rainfall model used in the random mixing are inferred from the synthetic data. I suppose that it is the case, but it should be clearly mentioned. If it is the case, it would be interesting to show the results of the fitting procedure. For instance: which copula (with which parameters) has been fitted? And also which marginal distribution? And how do the estimated values of the model parameters compare with the true ones (in this case you know the true values because it is a synthetic case)? In fact I suspect that the inferred statistical rainfall model cannot capture properly the true statistics of rainfall because the center of the rain cell is not observed. This can explain why conditional simulations (without streamflow constraints) cannot reproduce the observed hydrograph. I will come back to this point in my concerns about the method.

Yes the parameters are inferred from the synthetic data (only from the 'observations' though). Thus you are right, the inferred statistical model cannot capture properly the true statistics as for example the center of the rain cell is not observed. And this of course also leads to the fact that conditional simulations (without conditioning on runoff data) are not able to reproduce the observed hydrograph (but that is a general problem of course). The marginal distribution throughout the whole paper is the mixed distribution described in Eq.3 with a discrete probability of zeros ( $p_0$ ) and an

**HESSD**
exponential distribution for all values > 0. Based on the available observations the fitted parameters are:  $p_0 = 0.36$  and  $\lambda = 0.48$ . The fitted copula is a Gaussian copula with an exponential correlation function with a range of  $2.5 \ km$  in space and a range of  $1.5 \ hours$  in time.

Regarding the real world application. I would have been more convinced if you have shown an example with cross-validation. For instance the reconstruction of space-time rainfall for a well instrumented catchment (with many rain gauges). In this case you can select some stations for the inference of the mixing model parameters and the estimation of rain fields, and keep other stations to cross-validate the rain estimations. In addition, in the real world application, the altitude of the catchment ranges from 600m to 2500m; in this case one can expect some non-stationarities in rainfall statistics. Could you please discuss a bit this potential issue?

The presented real world application in this manuscript is more or less the initiator for this research. It is based on our multiyear research on hydrologic processes in this arid region under data scarcity and small scale rainstorms. We understand your wish for "... an example with cross-validation." and we acknowledge this idea. However, in this case data quality and situation is bad and scarce. The walnut gulch catchment in US might be more appropriate for an investigation with cross-validation, but not manageable now. We will consider this in our future research. Thank you for this hint. Regarding the non-stationarities in rainfall, in this case the application shows a reconstruction of a single rainstorm which doesn't consider rainfall non-stationarities. The Figure 1 below shows the measurements of the rainfall gauging stations for this event and their altitudes. Most of the rain is recorded on stations with lower altitudes located in the north-west and south-eastern part of the catchment. We will add this figure and information in the revised manuscript. Obviously much more research is needed to fully exploit the advantages and limits of this procedure but we thought that we are at a level so that results can be communicated to the advantage of the possible readers of the journal.
In the proposed method, the parameters of the hydrological model are supposed to be known and fixed. But at the same time the goal is to infer high resolution space-time rain fields to... improve hydrological modelling. This seems a bit circular reasoning. I see two options to break the circle:

- Either clearly acknowledge that the proposed framework is a first step that only aims at reconstructing space-time rain fields from rain gauge and streamflow measurements. Basically a proof of concept with strong assumptions (incl. known hydrological model), that will be relaxed only in future work. And in this case do not claim that the goal is to improve hydrological modelling, but just to show that doing reverse hydrology to reconstruct space-time rain fields is somehow feasible. In my opinion this is already a very nice contribution.

- Or improve the proposed framework to jointly reconstruct space-time rain fields and calibrate the hydrological model. This can be seen as the extension of the work of Del Giudice et al (2016) (see ref above) to the case of space-time rain fields. But I suspect that this will require a lot of developments... and I am not sure it will work in many configurations... But if it works it would be an even nicer contribution.

It is definitely option one and as you argued correctly, option two would require lots of developments and is not manageable within this manuscript. We had in mind that an improved estimate of the model input also improves the hydrologic modeling results. But you are right, this can be misunderstood and we will formulate our arguments more carefully in the revised manuscript.

In the synthetic case study, runoff simulations based on simulated rain fields conditioned to point observation only do not encompass the ones based on conditioning to rain gauge observations and streamflow observations. This is clearly visible by comparing figures 6 and 8. I am very surprised about it. Indeed, in my understanding,
the second case (adding conditioning to streamflow) should just add constraints to the first case. Therefore it should only select the rain fields obtained by simulation conditional to rain gauge only that are compatible with streamflow observations. But it is clearly not the case here... Therefore either I am missing something, and in this case I believe the reasons why this result arises must be explained in much more details by the authors; or there is some issue. One possible reason I could suggest (but I am not sure) is that the actual rain field that generates the observed hydrograph is kind of an extreme of the multivariate statistical distribution that underlies the random mixing model (after fitting model parameters). And therefore this extreme is not sampled by the 200 realizations performed in Figure 6.

Yes, your assumption is right and we are going to improve the discussion in the revised manuscript. Most probably, if we would sample more than 1000000 conditioned rainfall fields we would find a realisation which matches the runoff observation too, since the amount of possible conditioned rainfall fields is very much higher than the amount of rainfall fields matching point observation and runoff. Due to additional conditioning we find these realisation faster.

Regarding the assessment of uncertainty, I would be more cautious before stating "This ensemble can be used to describe the uncertainty in estimating spatio-temporal rainfall patterns" (p11, L9). In my opinion, the ensemble of realizations that is obtained is only a very partial descriptor of the total uncertainty. Indeed, in the proposed framework, both the statistical rainfall model and the runoff generation model have fixed parameters. Therefore the uncertainty originating from these two components is neglected. In the end, only the uncertainty related to the scarcity of the rain gauge measurement network is accounted for. I think this should be more clearly explained to the reader.

You are right. It is a partial descriptor of the total uncertainty. It describes the remaining uncertainty of spatio-temporal rainfall fields if all available data are exploited (under the assumption of known hydrologic model parameter, error-free measurements, and

**HESSD**
reliable statistical rainfall models). We believe that we can reduce the uncertainty of precipitation this way. We will formulate our arguments more carefully in the revised manuscript.

**Minor comments:**

-P2L22: In my knowledge, the turning band method is linked to the theory of random fields rather than to point processes.

To our knowledge, the turning band method has been introduced in its general form by Matheron (1973) and popularized for 2-D applications in hydrology by Mantoglou and Wilson (1982). So, it starts from 1-D point processes and was generalized to generate random fields.

-P2L34: "with respect to the outlined problem in the second paragraph above" - not clear what you are referring to. We will improve this.

-Modeling vs modelling: you have to choose one spelling. Thanks we will go with modeling in the revised manuscript.

-Eq 2: why qn and not q(t)? You are right, q(t) would make more sense. We'll change this in the revised manuscript.

-P4L20: It would be more clear if you say that P(x,t) is precipitation instead of rainfall. Or maybe call the variable R? That's also right, we will change it in the revised manuscript.
-Eq 3: Don't mix P and p.

This is actually correct but I admit that it is rather confusing. P represents a field, while p is a single value at a specific location within that field. However, the equations will be improved in the revised manuscript.

-Eq 4: Don't mix W and w. Same as for P and p.

-Eq 6: You should mention that conditioning is made at this step. Yes you are right we should point this out more clearly in the revised manuscript.

-Figure 3, 5, 7, 9: Please add units to the X and Y axes as well as to the color bar. You could also add the limits of the watershed. You are right. We will improve the figures.

-Real case study: you should show the observation dataset. We will add the figure shown here in the authors final comments.

**author's changes in manuscript**

Some hints regarding author's changes in manuscript have been already given in the comments section. Here, a summary of author's changes in manuscript based on comments of both referees is given.

• chapter 1 "Motivation": adding and discussion of literature regarding inverse hy-
drologic modeling, improved reasoning

- chapter 2.2 "Rainfall runoff model": description will be improved
- chapter 2.3 "Random Mixing for inverse hydrologic modeling": description will be improved
- chapter 3.2 "Results and discussion": discussion of the synthetic example will be enhanced and performed more precise.
- chapter 4.1 "Arid catchment test site": figure of rain gauge measurements and additional information will be added
- chapter 4.2 "Results and discussion" discussion of the real case study will be enhanced and performed more precise
- chapter 5 "Summary and conclusion" reasoning will be enhanced and performed more precise
- revision of Figures 3,5,7,9
- · all typos will be fixed and grammar will be improved
**HESSD**
Fig. 1. Rainfall amounts and altitudes of rainfall gauging stations for the case study area (or see supplement)

**Supplement:**

---

## Author Response (AR2)

**Authors reply to comments**

December 19, 2018

Journal: HESS
Title: Stochastic reconstruction of spatio-temporal rainfall pattern by inverse hydrologic modeling
Author(s): Jens Grundmann, Sebastian Hörning, András Bárdossy
MS No.: hess-2018-350

We would like to thank the editor and the referees for their time to review the manuscript again. Our reply is organized as follows: (1) comments from Editor and Referees are in black color, (2) author's response is marked in blue color and placed within referee comments whenever it's needed, (3) author's changes in manuscript are highlighted within author's responses followed by a marked-up manuscript with tracked changes.

**Editor comments**

Minor comments [page line]
[2 7] Isn't (iii) similar to (i)?
Also (ii) can be similar for special cases. The paragraph addresses the challenges of traditional interpolation methods applied for (or within) distributed hydrologic models from different perspectives: (i) measuring network density (ii) rainstorm characteristics (iii) partly covered catchments, which tend to smoothed pattern in areal rainfall.
[2 10] "where rainstorms show a great variability in space and time".Please add a reference to support this statement.
Reference is given at the end of the sentence and fits all statements within this sentence (Pilgrim1988).
[2 16] rader.
Done.
[2 19] "A large amount of literature exists describing different approaches for space-time simulation of rainfall fields..." - I suggest also referencing here more recent stochastic rainfall generator models.
Some references are given at the end of the paragraph. We add: Peleg etal. 2017: "An advanced stochastic weather generator for simulating 2-D high-resolution climate variables" (Please note, we didn't want to write a review paper.)
[3 8] "three-dimensional rainfall fields". space and time?
We removed "... three-dimensional ... " since explanation is given afterwards. Space and time is right, but 3D refers to the technical realisation by 3D matrix with two dimensions in space and one in time.
[3 16] Replace "chapter" with "section".
Done.
[3 19] "as well as conditional rainfall simulations only".not clear, please rephrase.
We write now: "A synthetic test site is introduced which is used to demonstrate and discuss: (i) the limits of common hydrologic modeling approaches (using rainfall interpolation), and (ii) the shortcomings of rainfall simulations which are not conditioned on the observed runoff.
[3 30] "In order to additionally condition the rainfall field on the observed runoff it is iteratively updated". Please rephrased this sentence.
We rephrase the sentence: "Afterwards, an optimization is performed to additionally condition the rainfall field on the observed runoff."
[4 13] not Pe(x,y,t)? Please indicate what are x and t in the text.
We add: "... with location $x \in D$ and time $t \in T$ ..."

[4 14] Ia - for each grid cell? should be - Ia(x,y)?

We change it to $I_a(x)$, $f_c(x)$, $f_r(x)$

[4 25] Up to now... ?

Removed.

[4 28-29] Repetitive.

We removed "... first presented in Bárdossy and Hörning (2016a) and Bárdossy and Hörning (2016b) where the authors have applied it to inverse groundwater modeling problems."

[5 10] "After identifying the observations" - identifying how?

We write now: "Using the given observations ..."

[8 20] The spatial extension is fixed in time?

No, it is not fixed. It develops over time. We write now: "... a maximum spatial extension ..." .

[8 22] Figure 3 is for the total rainfall over the event or a snapshot in time? Is the synthetic storm advects with time?

Figure 3 is for the total rainfall over the event. The synthetic storm doesn't advect with time. We add an additional figure in the supplementary material showing the development over time and expand the figure reference. We write now: "... (see Figure 3 and Figure A1 in supplementary material)."

[Figure 2] The North arrow in the figure can be deleted, as this is a synthetic example

Done.

[A general comment] Please use decimal point instead of a comma (i.e. replace 0,36 with 0.36)

Done.

[12 2] Why 107? And later [16 2] – why 58?

We wanted to have a sample size larger than 100 with NSE > 0.7 and let the algorithm do its job. After that, a refinement was carried out regarding NSE value. 58 is the sample size after refinement for the real world application. To clarify this we write now: "... an ensemble of 58 different hydrographs is obtained after refinement ..." instead of "...an ensemble of 58 different hydrographs is simulated ..."

[Figure 13] I suggest moving this figure to the supplementary material.

We like to keep this figure since it explains the behaviour in runoff.

[12 26] cell wise?

We write now: "...mean value per grid cell ..."

[Section 4.2] I suggest to show here examples of how the total rainfall of this event was spatially distributed (3-5 best members) in comparison to the interpolated total rainfall field from the rain gauges.

We provide this figures in the supplementary material since there are already a lot of figures in the manuscript. In our opinion the figure doesn't support the explanation in runoff behaviour very well, but it underlines why figure 13 is required. We hope you can follow our arguments. We expand the figure reference and write now: "(Figure 13, see also Figure A2 in supplementary material for comparison of event based spatial rainfall amounts)"

**Anonymous Referee #2**

Suggestions for revision or reasons for rejection (will be published if the paper is accepted for final publication) The revisions performed by the authors significantly improved the original manuscript. In my opinion it is now almost ready for publication, although the written English still needs to be improved. But I think this can be handled during the proof stage. I also have some very minor comments about the content of the paper, but I think they can be implemented at the proofs stage as well (if the authors find them relevant).

* It would be great if the source codes of both the hydrological model and the Random Mixing model would be made freely available. It can help interested readers to better understand the method, and interested practitioners to use it.

They are available on request. Currently, they are not in a shape that we can publish them.

* I think you should emphasize a bit more on the fact that the use of Random Mixing in the present context allows to perform the hydrological inversion by optimization and not through sampling. You already say it indirectly (e.g. p4, l30 and p17, l15-16) but it could be more clear, because it is a very nice feature of the proposed framework.

Thanks for this hint. We rephrase p3,l30: "Afterwards, an optimization is performed to additionally condition the rainfall field on the observed runoff."

We rephrase p17,l15-16: "By applying optimization, rainfall fields are conditioned on discharge too, and appropriate candidates ..." And we add a sentence in the following paragraph and write now: "The inference of a three dimensional input variable by using an integral output response results in a set of possible solutions in terms of spatio-temporal rainfall pattern. This ensemble is obtained by repetitive execution of the optimization step within the Monte-Carlo loop. It can be considered as a descriptor of the partial uncertainty ..."

* P2, l35: 'stochastic rainfall simulations [. . . ] can fail at matching the observed stream flow if rainfall fields are conditioned on rainfall point observation only.' -> A reference would be interesting (even if you prove it in the following of the paper).

To my knowledge, there is no paper showing this explicitly. Casper etal. 2009 indicates this, but the scope of the study was to investigate the capability of extrapolation for different hydrologic models.

* P4, 2.2 Rainfall runoff model -> I still miss some equations to sum up the description of the model you give from line 12 to line 26.

We would like to keep it as it is.

* P7, l22: $\ll 1$ -> You could specify how small this sum must be to ensure correct results.

The results are correct as soon as this sum is smaller than 1. We only want it to be much smaller than 1 because that makes the field W* very smooth and with that the fields Uk have more impact.

* P9, l1-2: you mention the inferred values of your rainfall model parameters (i.e. range=2.5km, etc.) but you do not mention the actual values used to generate the reference field. It would be interesting to mention them for comparison.

The synthetic reference field wasn't generated on the basis of copulas, but we determined the parameter by using the full synthetic dataset. The range in space is 4.5km and 2.5 h in time. We incorporate this information in the manuscript and add a sentence at the end of section 3.1. : ..."In comparison, using the full synthetic dataset a range of 4.5 km in space and a range of 2.5 h in time are estimated."

* P9, l2: the range in time should be 1.5h and not 1.5km.

Done.

*p12, l10: 'the rainfall event can be localized and reconstructed in its spatial extent as well as its course in time' -> you do not show how good is the reconstruction over time (you only show event based averaged rain field reconstructions).

We provide an additional figure for the supplementary material which shows the course in time of the "observation" in comparison to two selected realisations. We expand the figure reference in section 3.1 and write now: "... (see Figure 3 and Figure A1 in supplementary material)." We add a figure reference in section 3.2.3 in the first paragraph after "... its course in time". We add: "(see also Figure A1 in supplementary material). "

* p13, l5: You could maybe mention that this case study was the starting point for the present study (as you mentioned in the response to my first review).

Thanks for this hint. We add: "It is the starting point for the present study and part of our multiyear research on hydrologic processes in this region." after the first sentence in section 4.1

*p15, l2: the range in time should be 1h and not 1km.

Done.

**Anonymous Referee #1**

Suggestions for revision or reasons for rejection (will be published if the paper is accepted for final publication) The authors have revised the paper and addressed my many confusions from their first draft. The logic they are proposing is now quite understandable. My only suggestions are at this stage to enhance their discussion of the literature to acknowledge other papers that have tried to capture spatio-temporal uncertainty in rainfall inputs along the lines of what is being attempetd here. A paper that I can think of is: Abu Shoaib, S., L. Marshall, and A. Sharma (2016), A metric for attributing variability in modelled streamflows, Journal of Hydrology, 541, 1475-1487, doi:http://doi.org/10.1016/j.jhydrol.2016.08.050. - it quantifies the impact of uncertainty in precipitation inputs on modelled flows by considering alternate sources of uncertainty.

Thanks for this hint. The paper is very interesting but it doesn't fit directly to our manuscript. Input or rainfall uncertainty is not explicitly addressed and it contains no inverse modeling approach. It's a straightforward hydrologic modeling study trying to distinguish between different sources of uncertainties

(model structure, model parameters, objective function, and their interaction). I have absolutely no idea where to place this paper. Maybe in the outlook section in the last paragraph, but there would fit lots of other papers too. We will consider it for our future work.

Other than that, I would have preferred that the authors consider a more detailed real world example as they have mentioned in their discussion, and show the clear applicability of their approach in other settings. However, I leave whether this should be included as part of this presentation or a future more enhanced presentation a decision for the editors to take.

[revised manuscript text omitted]